# Opt-Miner: Empowering Information-Seeking Agent with Tree-Guided Data Synthesis for Optimization Modeling

**Haoyang Liu** [1][*]  **Yuyang Cai** [1][*]  **Jie Wang** [1][†]  **Xiongwei Han** [2]  **Minyang Hu** [2]  **Shuqi Liu** [2]  **Mingxuan Yuan** [2]
**Jianye Hao** [2][3]  **Feng Wu** [1]

## Abstract

Large Language Model (LLM) agents have shown significant potential in automated optimization modeling for mathematical problems. However, real-world problems are still challenging due to their knowledge-intensive nature. Existing methods, constrained by static parametric knowledge, often lack the domain expertise required to comprehend complex scenarios and apply appropriate mathematical techniques, leading to errors. To address this challenge, we propose the Opt-Miner framework, where the agent learns to identify missing knowledge, retrieve technical documents on the web, and ground its mathematical models for improved modeling performance. The core of Opt-Miner is a novel tree-guided data synthesis pipeline coupled with a retrieval-based group relative policy optimization (R-GRPO) algorithm, designed to foster the agent's information-seeking capabilities. Specifically, we first formulate each problem into a tree structure, with its scenario contexts and mathematical techniques embedded in subtrees. We then employ subtree union, transfer, and knowledge fogging to synthesize complex, multi-domain problems that incorporate knowledge gaps, thereby necessitating active information seeking to solve these problems. Based on synthesized data, we propose R-GRPO for agent reinforcement learning. Experiments demonstrate that Opt-Miner-Qwen3-8B achieves performance comparable to 32B state-of-the-art specialized agents and commercial reasoning models.

---

[*]Equal contribution . This work was completed while Haoyang Liu was an intern at Noah's Ark Lab, Huawei Technologies. Haoyang Liu <dgyoung@mail.ustc.edu.cn>. [†]Corresponding author. [1] MoE Key Laboratory of Brain-inspired Intelligent Perception and Cognition, University of Science and Technology of China [2] Noah's Ark Lab, Huawei Technologies [3] Tianjin University . Correspondence to: Jie Wang <jiewangx@ustc.edu.cn>.

*Proceedings of the 43rd International Conference on Machine Learning*, Seoul, South Korea. PMLR 306, 2026. Copyright 2026 by the author(s).

## 1. Introduction

Mathematical modeling and Operations Research (OR) are essential tools for solving a wide range of mathematical problems in modern industries such as factory scheduling (Muckstadt & Wilson, 1968), logistics planning (Pochet & Wolsey, 2006; Bai et al., 2025), and energy management (Moreno et al., 2015). In practice, translating a natural language problem description into a rigorous mathematical optimization model is complicated. This process typically necessitates extensive, iterative dialogues between domain experts and stakeholders to uncover nuanced, industry-specific constraints——a cycle that is inherently time-consuming and prone to inefficiency when tackling unfamiliar domains. Considering such a labor-intensive bottleneck, large language models (LLMs) have emerged as a promising solution, offering the potential to automate the transformation of unstructured problem descriptions into precise mathematical formulations (Xiao et al., 2024; Huang et al., 2025).

Recent advancements in autonomous mathematical modeling generally fall into two categories: prompt-based and fine-tuning-based methods. Prompt-based approaches leverage pretrained LLMs through sophisticated pipelines, including multi-agent systems (Xiao et al., 2024; Ahmaditeshnizi et al., 2024), Monte Carlo Tree Search (Astorga et al., 2025), and specialized retrieval modules (Liu et al., 2025). The fixed reasoning workflows inherent in these approaches restrict their adaptability to more complex problem structures. Conversely, fine-tuning-based methods focus on enhancing the reasoning capabilities by training the model on synthesized data, as seen in ORLM (Huang et al., 2025), LLMOPT (Jiang et al., 2025), OptMATH (Lu et al., 2025), and SIRL (Chen et al., 2026). Although these methods generally yield better performance on standard benchmarks, the long-tail nature of industrial optimization makes it practically impossible to internalize all domain expertise within static model parameters. This restricts their ability to generalize to unseen situations or heterogeneous domains, due to significant knowledge gaps in problem scenario comprehension and mathematical modeling techniques.

Thus, we argue that agents must evolve from passive reasoners into active information seekers capable of exploring

external professional knowledge on the web. For model training, we identify two drawbacks: (1) Existing training datasets feature single-domain, self-contained problems where scenario comprehension is trivial, failing to provide a meaningful incentive for information-seeking. (2) There is a lack of supervised signals for the information-seeking trajectory. Relying solely on final modeling accuracy fails to credit the quality of intermediate retrieval, leaving the agent unable to distinguish effective information seeking from redundant exploration. Consequently, we need to synthesize problems that densely integrate multi-domain knowledge while providing explicit ground-truth retrieval evidence.

To this end, we propose Opt-Miner, an agentic information-seeking framework tailored to bridge the gap in scenario comprehension and mathematical techniques. Specifically, we first introduce a data synthesis pipeline designed to cultivate and train an agent's web search capability. We reformulate each modeling problem into a hierarchical scenario tree (HST), which embeds the problem scenario contexts and relevant mathematical techniques (such as big-M methods and other transformations (Pu et al., 2024)) in its subtrees. By employing subtree union, transfer, and knowledge fogging operators, we synthesize complex, multi-domain problems that incorporate intentional knowledge gaps with ground-truth retrieval evidence, thereby compelling the agent to engage in active information seeking. For training, we optimize the agent using our proposed retrieval-based group relative policy optimization (R-GRPO). By incorporating a hybrid reward that evaluates both modeling accuracy and intermediate retrieval quality, Opt-Miner learns to proactively identify missing knowledge, retrieve external documents, and ground its mathematical formulations.

Experimental results demonstrate that Opt-Miner significantly improves over 20% modeling accuracy across various benchmarks, particularly in difficult ones where internal model knowledge is insufficient to ensure a sound formulation. With our generated data, Opt-Miner-Qwen3-8B achieves performance comparable to 32B state-of-the-art specialized agents and commercial reasoning models.

## 2. Related Work

**LLM for Mathematical Modeling Problem** Machine Learning has been widely applied to Operations Research (Bengio et al., 2021). While early works focus on accelerating solving NP-hard problems (Pu et al., 2026), LLM-based optimization modeling has emerged as a promising approach to reducing the technical barriers and time costs of mathematical modeling tasks. Existing methods in this field fall into two main categories: prompt-based workflows and fine-tuned specialized models. Representative prompt-based works such as CoE (Xiao et al., 2024), Opti-MUS (Ahmaditeshnizi et al., 2024) and Opt-Verifier (Liu

et al., 2026a) adopt multi-agent collaboration to iteratively generate solver codes, while others leverage search-based techniques like MCTS (Astorga et al., 2025) for exploring model component spaces or retrieval techniques like OptiTree (Liu et al., 2025). On the other hand, fine-tuned models including ORLM (Huang et al., 2025), LLMOPT (Jiang et al., 2025), OptMATH (Lu et al., 2025) and MIND (Liu et al., 2026b) rely on large-scale datasets for training. In terms of training data synthesis, ORLM (Huang et al., 2025) proposes the OR-Instruct3K dataset, and OptMATH (Lu et al., 2025) introduces a novel data synthesis framework to generate massive training data. However, existing datasets are not designed for search-oriented tasks and thus lack the essential components required to train agents in proactive knowledge retrieval and the utilization of external information for optimization modeling.

**Autonomous Information Seeking** To fill knowledge gaps in tackling complex, knowledge-intensive tasks, researchers have enabled LLMs to autonomously retrieve rich, cross-domain information from the web (Xu & Peng, 2025). Early studies have validated the effectiveness of integrating search modules for reasoning and information retrieval, with representative systems including R1-Searcher (Song et al., 2025), Search-o1 (Li et al., 2025), and Simple Deep-Researcher (Sun et al., 2025). For model training, state-of-the-art training or data synthesis approaches have been adopted, such as model alignment (Li et al., 2026), duplicate sampling policy optimization (Li et al., 2025), hybrid reward systems (Tao et al., 2026), and energy-guided reinforcement learning (Dong et al., 2026). For training algorithms, existing methods mainly use DPO (Rafailov et al., 2023; Liang et al., 2025) or GRPO (Shao et al., 2024).

## 3. Preliminaries: Formalization of Information Seeking for Optimization Modeling

We formalize the modeling of knowledge-intensive optimization problems as a sequential decision-making process. Given a problem $\mathcal{P}$, the agent iteratively performs reasoning and information seeking before writing the final solver code to produce the answer. We maintainin a trajectory history $\mathcal{H}_{t-1} = \{P, (\tau_1, a_1, o_1), \ldots, (\tau_{t-1}, a_{t-1}, o_{t-1})\}$ where $\tau_i$, $a_i$, and $o_i$ denote reasoning thoughts, actions, and observations, respectively. At each research step $t \in \{1, \ldots, T\}$, the agent first generates a reasoning thought $\tau_t$, followed by an action $a_t$ selected from a toolset $\mathcal{A} = \{\text{Web Search Query}, \text{Python Code Execute}\}$. The agent then receives an observation $o_t$, which may be a summarized technical document from the web search or an execution log from the Python interpreter. This iterative cycle continues until the agent determines it has gathered sufficient evidence, where it generates the definitive solver code to obtain the final answer $y$. The resulting trajectory is

Problem Scenario Tree $\mathcal{T}$

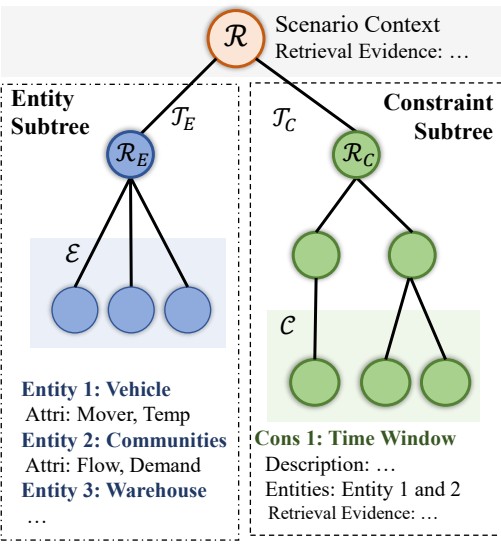

*Figure 1.* Illustration of the hierarchical scenario tree.

formalized as

$$\mathcal{H}_\mathcal{P} = \{(\tau_1, a_1, o_1), \ldots, (\tau_T, a_T, o_T), \text{Code}, y\}. \quad (1)$$

# 4. Opt-Miner: Agentic Information-Seeking Framework for Optimization Modeling

In this section, we present Opt-Miner, an agentic information-seeking framework tailored for optimization modeling tasks. In Section 4.1, we introduce the data synthesis pipeline designed to stimulate the agent's information-seeking behavior. In Section 4.2, we detail the training pipeline of the agents.

## 4.1. Tree-Guided Data Synthesis Pipeline

We introduce the data synthesis pipeline. To analyze the scenario structure of optimization problems, we formulate a problem $\mathcal{P}$ as a Hierarchical Scenario Tree (HST) $\mathcal{T}$ in Section 4.1.1. We then present the similarity metrics and the subtree evolution operators in Section 4.1.2. Finally, we present the data validation process in Section 4.1.3.

### 4.1.1. HIERARCHICAL SCENARIO TREE

To analyze the scenario and mathematical structure of problems, we formulate a problem $\mathcal{P}$ as a Hierarchical Scenario Tree (HST) $\mathcal{T}$. The root node $\mathcal{R}$ represents the global scenario context and high-level industrial background. To embed structured scenario information and mathematical techniques, the tree branches into two sub-structures: the entity subtree $\mathcal{T}_E$, which contains the subject entities in the problem, and the constraint subtree $\mathcal{T}_C$, which organizes the conditions and corresponding mathematical modeling

techniques. We present an example of HST in Figure 1.

**The Entity Subtree** $\mathcal{T}_E$ captures the subject entities—such as machinery, fleets, or inventory in the problem—which serve as the foundation for decision variables and parameters. Rooted at $\mathcal{R}_E$, each child node represents a specific entity. A key insight is that diverse entities across different domains often share identical functional roles; for instance, packages and containers both function as storage units, while vehicles and conveyors act as movers. To leverage this, we employ a predefined candidate set of high-level attributes and utilize large language models to map these attributes to entities based on their roles. This abstraction enables the framework to uncover similarities across different problem domains for multi-domain scenario synthesis.

**The Constraint Subtree** $\mathcal{T}_C$ categorizes problem mathematical constraints into thematic groups. Under the root node $\mathcal{R}_C$, the tree branches into constraint clusters, each representing a high-level category such as operational safety, capacity limits, or resource budgets. Within these clusters, the child nodes define specific constraints. Each constraint node contains a natural language description, links to the associated entities in $\mathcal{T}_E$, and the required mathematical techniques to formulate this constraint.

### 4.1.2. TREE EVOLVING FOR CHALLENGING PROBLEMS

To construct multi-domain, complex problems, we evolve seed problems into complex scenarios. We first retrieve problem descriptions and mathematical models from arXiv using keywords related to various industrial domains. These collected instances serve as seed problems to build a library of base HSTs. Crucially, we record the associated search keywords in the root node, and the corresponding mathematical modeling techniques in the constraint nodes, constructing the ground-truth retrieval evidence $\mathcal{Q}$ for the agent. To systematically enhance problem diversity and complexity, we employ a two-stage evolution process: similarity identification followed by tree evolution through subtree union, transfer, and knowledge fogging operators.

**Similarity Identification.** The basis for effective tree evolution is identifying compatible HSTs. We define two metrics to evaluate the similarity between trees $\mathcal{T}_i$ and $\mathcal{T}_j$: **thematic compatibility** $S_{\text{sem}}$ and **entity functional compatibility** $S_{\text{func}}$. The first metric, $S_{\text{sem}}$, evaluates whether two problems belong to the same problem domain, such as logistics or manufacturing. This is computed using an LLM-based classifier with a binary output:

$$S_{\text{sem}}(\mathcal{T}_i, \mathcal{T}_j) = \text{LLM}(\text{Prompt}_{\text{cmp}}, \mathcal{R}_i, \mathcal{R}_j) \quad (2)$$

where $\mathcal{R}$ represents the root node containing the global problem scenario contexts and $\text{Prompt}_{\text{cmp}}$ is a structured comparison prompt. The second metric, $S_{\text{func}}$, measures

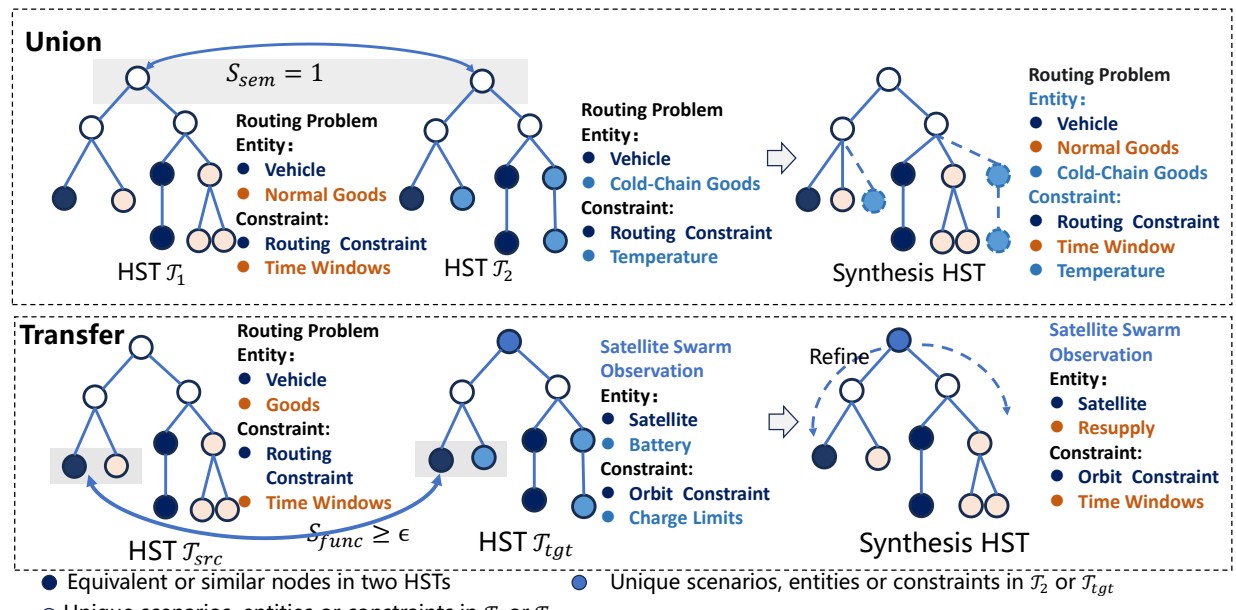

*Figure 2.* Overview of the Opt-Miner Data Synthesis Pipeline. (Top) Scenario Union merges compatible HSTs to create complex problems. (Bottom) Scenario Transfer adapts mathematical structures across heterogeneous domains.

the similarity based on entity attributes. Let $\mathcal{E}$ be the set of entities in the subtree $\mathcal{T}_E$, and $\mathcal{A}(e)$ be the set of attributes of entity $e$. We define the similarity of two entities $(e_m, e_n)$ using the attribute overlap ratio $|\mathcal{A}(e_m) \cap \mathcal{A}(e_n)|/|\mathcal{A}(e_m)|$. Crucially, we define two entities as functionally equivalent if this ratio is 1. Finally, the functional compatibility of two problems is calculated as the average maximum ratio:

$$S_{\text{func}}(\mathcal{T}_i, \mathcal{T}_j) = \frac{1}{|\mathcal{E}_i|} \sum_{e_m \in \mathcal{E}_i} \max_{e_n \in \mathcal{E}_j} \frac{|\mathcal{A}(e_m) \cap \mathcal{A}(e_n)|}{|\mathcal{A}(e_m)|}. \quad (3)$$

This metric enables the framework to identify functional commonalities across domains. For instance, a warehouse storage rack and a factory inventory shelf are identified as functionally equivalent because they share the attributes of "container with storage capacity." We consider a pair $(\mathcal{T}_i, \mathcal{T}_j)$ as a candidate for evolution if either $S_{\text{sem}} = 1$ or $S_{\text{func}} \geq \epsilon$ exceeds a predefined threshold, ensuring the generated scenarios are logically consistent.

**Operator I: Scenario Union via Subtree Merging.** This operator merges problems within the same or closely related domains. For scenario pairs $(\mathcal{T}_i, \mathcal{T}_j)$ with high thematic compatibility ($S_{\text{sem}} = 1$), we invoke the scenario union operator to create a unified problem $\mathcal{P}_{\text{syn}}$. The first step is merging the entity subtrees. Based on the definition, any functionally equivalent entities in $\mathcal{T}_j$ and $\mathcal{T}_i$ is treated as a duplicate and merged into a single entity node. Unique entities from both sides are preserved, forming the synthesized entity set $\mathcal{E}_{\text{syn}}$. Since $S_{\text{sem}} = 1$, the newly added entities naturally fit into the problem scenario context of $\mathcal{T}_i$. Next, we aggregate the

constraint clusters to form $\mathcal{C}_{\text{syn}} = \mathcal{C}_i \cup \mathcal{C}_j$. Finally, we use a large language model to weave these elements into a single, cohesive problem description:

$$\mathcal{P}_{\text{syn}} = \text{LLM}(\text{Prompt}_{\text{syn}}, \mathcal{R}_i, \mathcal{E}_{\text{syn}}, \mathcal{C}_{\text{syn}}). \quad (4)$$

This operator simulates the integration of complex industrial systems, such as merging two independent warehouse scheduling tasks into a unified cross-warehouse problem.

**Operator II: Scenario Transfer and Entity Adaptation.** To generalize the modeling logic and techniques across heterogeneous domains, we define the scenario transfer operator. This operator is applied to pairs $(\mathcal{T}_{src}, \mathcal{T}_{tgt})$ that belong to different domains ($S_{\text{sem}} = 0$) but share high functional compatibility $S_{\text{func}} \geq \epsilon$. The goal is to transplant the mathematical structure of the source problem into the the target domain. First, we adapt the source entities $\mathcal{E}_{src}$ to fit the target problem scenario context $\mathcal{R}_{tgt}$. If a source entity lacks a direct equivalent in the target problem $\mathcal{T}_{tgt}$, the LLM rewrites it into a new entity that aligns with the target's scenario while preserving its original entity attributes. Then, the constraint subtree $\mathcal{C}_{src}$ is applied to these adapted entities. This process generates a new problem,

$$\mathcal{P}_{\text{syn}} = \text{LLM}(\text{Prompt}_{\text{trans}}, \mathcal{R}_{tgt}, \mathcal{E}_{src}, \mathcal{C}_{src}). \quad (5)$$

This operator forces the agent to perform deep structural comprehension rather than relying on surface-level memorization of domain-specific terms.

**Operator III: Knowledge Fogging.** To simulate real-world scenarios where information is often implied rather

than explicit, we introduce the knowledge fogging operator. Standard training datasets typically provide self-contained problems where most conditions are fully explained. Consequently, models develop a false sense of confidence in the sufficiency of the provided context, learning to rely solely on the given problem description and ignoring the need for external information. To counter this, our operator targets the problem scenario contexts and constraint clusters $\mathcal{C}$ that involve widely recognized industrial standards or mathematical principles. We employ an LLM to rewrite these details, specifically masking the explicit domain common knowledge or replacing them with concise professional terminology. Formally, we generate an abstracted version $\mathcal{C}_{\text{abs}}$:

$$\mathcal{C}_{\text{abs}} = \text{LLM}(\text{Prompt}_{\text{abs}}, \mathcal{C}). \tag{6}$$

For instance, a detailed constraint explaining that "*inventory at the current step equals the previous inventory plus production minus demand*" is masked as implicit conditions. This compels the agent to autonomously identify the underlying mathematical formulation (e.g., $I_t = I_{t-1} + P_t - D_t$), rather than simply translating the provided text.

### 4.1.3. DATA FILTERING AND VALIDATION

To ensure the quality of the synthesized data, we implement a rigorous filtering pipeline. Drawing on the solver-informed verification methodology from Chen et al. (2026), we adopt a consistency check mechanism that goes beyond simple majority voting. We retain only those problems where multiple independent trials yield identical optimal objective values and consistent model structures. This approach effectively filters out infeasible or logically inconsistent cases. For the construction of our evaluation benchmark, named Opt-Miner-Bench, we further incorporate a manual review and correction process conducted by human experts to ensure high data reliability.

### 4.2. Agentic Training Framework

Building upon the synthesized data, we present the agentic training framework designed to cultivate the agent's information-seeking capabilities. The core of this framework is a hybrid reward system that provides supervision signals for both the quality of the information-seeking process and the accuracy of the final optimization models.

**Hybrid Reward System.** We propose a hybrid reward function to evaluate the agent's performance. Let $\mathcal{P}$ be an optimization problem with a ground-truth answer $y^*$ and the ground-truth retrieval evidence $\mathcal{Q}$, which is the set of essential keywords and mathematical techniques recorded during data synthesis. During the information seeking process, the agent generates a set of search queries. Let $\tilde{\mathcal{Q}}$ denote the set of keywords extracted from these queries. To evaluate the research quality, we define the retrieval matching reward $R_{\text{ret}}$ as the F1-score between the agent's search terms and the ground-truth evidence $\text{Pre}_{\text{ret}} = |\mathcal{Q} \cap \tilde{\mathcal{Q}}|/|\mathcal{Q}|, \quad \text{Rec}_{\text{ret}} = |\mathcal{Q} \cap \tilde{\mathcal{Q}}|/|\tilde{\mathcal{Q}}|,$

$$R_{\text{ret}} = 2 \cdot \frac{\text{Pre}_{\text{ret}} \cdot \text{Rec}_{\text{ret}}}{\text{Pre}_{\text{ret}} + \text{Rec}_{\text{ret}}}. \tag{7}$$

A high $R_{\text{ret}}$ indicates that the agent has accurately identified the necessary knowledge without engaging in excessive or redundant exploration.

The final modeling performance is evaluated by the modeling accuracy reward $R_{\text{acc}}$, which relies on a solver-based verification of the generated code. Let $y$ be the optimal objective value obtained from the generated model and $y^*$ be the ground-truth value. The reward is formulated as $R_{\text{acc}} = \mathbb{I}(|y - y^*| < \alpha)$, where $\mathbb{I}(\cdot)$ is the indicator function and $\epsilon$ is a numerical tolerance. The total reward is computed as a weighted sum of the accuracy, retrieval, and a basic format reward $R_{\text{format}}$ (which ensures the output follows the required structure):

$$R_{\text{total}} = \omega_a R_{\text{acc}} + \omega_r R_{\text{ret}} + R_{\text{format}}. \tag{8}$$

**Retrieval-based Group Relative Policy Optimization** We optimize the agent's policy $\pi_\theta$ using our proposed Retrieval-based Group Relative Policy Optimization (R-GRPO). This algorithm adapts the standard GRPO framework (Shao et al., 2024) to the context of our multi-turn information-seeking process. Specifically, for each problem instance $\mathcal{P}$, we sample a group of $G$ trajectories $\{\mathcal{H}_1, \ldots, \mathcal{H}_G\}$ from the current policy. We then compute the relative advantage $\hat{A}_i$ for the $i$-th trajectory by normalizing its cumulative hybrid reward $R_{\text{total},i}$ against the group statistics:

$$\hat{A}_i = \frac{R_{\text{total},i} - \frac{1}{G}\sum_{j=1}^{G} R_{\text{total},j}}{\text{std}(\{R_{\text{total},j}\}_{j=1}^{G}) + \eta} \tag{9}$$

where $\eta$ is a small constant for numerical stability. The policy is updated by minimizing a clipped surrogate objective, incorporating a token-level masking strategy to strictly supervise agent-generated content:

$$\mathcal{L}(\theta) = -\mathbb{E}_{\mathcal{P},\{\mathcal{H}_i\}}\left[\frac{1}{G}\sum_{i=1}^{G}\frac{1}{|\mathcal{H}_i|}\sum_{t=1}^{|\mathcal{H}_i|}\mathbb{M}_t \min\left(r_{i,t}(\theta)\hat{A}_i,\right.\right.$$

$$\left.\left.\text{clip}(r_{i,t}(\theta), 1-\epsilon, 1+\epsilon)\hat{A}_i\right) - \beta\mathbb{D}_{KL}(\pi_\theta||\pi_{\text{ref}})\right] \tag{10}$$

In this formulation, $r_{i,t}(\theta)$ denotes the importance sampling ratio $\pi_\theta(y_{i,t}|\cdot)/\pi_{\theta_{\text{old}}}(y_{i,t}|\cdot)$, and $\mathbb{D}_{KL}$ represents the Kullback-Leibler divergence from the reference policy $\pi_{\text{ref}}$. Crucially, we apply a binary mask $\mathbb{M}_t$ to distinguish trainable outputs from environmental inputs. We set $\mathbb{M}_t = 1$ for

*Table 1.* Performance comparison on optimization modeling benchmarks. Scores represent the Pass@1 accuracy (%). Bold indicates the best performance among the finetuned modeling agents, and the underline indicates the second best. The star * means that the data are cited from other papers (except for Opt-Miner-Bench).

| Method | NL4Opt | MAMO | | IndustryOR | Resocratic | OptMATH | Opt-Miner-Bench |
| | | EasyLP | ComplexLP | | | | |
| --- | --- | --- | --- | --- | --- | --- | --- |
| *Commercial LLM Agents* | | | | | | | |
| GPT-4o* | 79.2 | 81.3 | 41.2 | 38.0 | 48.4 | 17.5 | 15.6 |
| OpenAI-o3* | 86.8 | 79.2 | 53.5 | 43.0 | 62.8 | 43.3 | 25.0 |
| *Open-Source LLM Agents* | | | | | | | |
| DeepSeek-V3 | 79.8 | 95.2 | 53.2 | 36.0 | 55.1 | 23.4 | 23.4 |
| DeepSeek-R1 | 82.4 | 87.2 | 67.9 | 45.0 | 67.2 | 34.9 | 19.5 |
| Qwen3-4B-Instruct | 74.1 | 65.3 | 15.2 | 19.0 | 40.8 | 22.9 | 5.5 |
| Qwen3-8B | 77.5 | 74.6 | 21.3 | 24.0 | 43.7 | 16.2 | 13.3 |
| *Prompt-based Modeling Agents* | | | | | | | |
| CoE* | 64.2 | - | 40.2 | - | - | - | - |
| OptiMUS* | 78.8 | 77.0 | 43.6 | 31.0 | 45.8 | 20.2 | - |
| AutoFormulation* | 92.6 | - | 62.3 | 48.0 | - | - | - |
| *Finetuned Modeling Agents* | | | | | | | |
| ORLM-Llama3-8B* | 85.7 | 82.3 | 37.4 | 38.0 | 51.1 | 2.6 | 3.9 |
| Step-Opt-Llama3-8B* | 84.5 | 85.3 | 61.6 | 36.4 | - | - | - |
| LLMOPT-Qwen2.5-14B* | 80.3 | 89.5 | 44.1 | 29.0 | 53.8 | 12.5 | 7.0 |
| OptMATH-Qwen2.5-7B* | 94.7 | 86.5 | 51.2 | 20.0 | 57.9 | 24.4 | 10.1 |
| OptMATH-Qwen2.5-32B* | 95.9 | 89.9 | 54.1 | 31.0 | 66.1 | 34.7 | 17.2 |
| SIRL-Qwen2.5-7B* | 96.3 | 91.7 | 51.7 | 33.0 | 58.0 | 30.5 | 12.5 |
| SIRL-Qwen2.5-32B* | **98.0** | 94.6 | 61.1 | 42.0 | **67.4** | **45.8** | 21.9 |
| Opt-Miner-Qwen3-4B-Instruct | 96.2 | 92.5 | 66.4 | 41.0 | 60.6 | 35.5 | 21.1 |
| Opt-Miner-Qwen3-8B | 97.0 | **94.8** | **76.5** | **44.0** | 66.5 | **45.8** | **30.5** |

tokens corresponding to the agent's actions—specifically reasoning thoughts $\tau_t$, search queries $q_t$, and the final solver code—while excluding tokens derived from external document observations $d_t$ (i.e., $\mathbb{M}_t = 0$).

# 5. Experiments

## 5.1. Experiment Setups

**Dataset** We evaluate the effectiveness of Opt-Miner across seven diverse modeling benchmarks, covering a wide range of difficulty levels and problem domains. These datasets include NL4Opt (Ramamonjison et al., 2021) for elementary-level linear programming, MAMO (Huang et al., 2024) (with both EasyLP and ComplexLP subsets), IndustryOR (Huang et al., 2025) featuring real-world industrial cases, Resocratic (Yang et al., 2025), and the challenging OptMATH benchmark (Lu et al., 2025). Opt-Miner-Bench is our proposed benchmark with 128 problems, constructed via our data synthesis pipeline with human expert inspection. Due to the limited space, we present a detailed analysis of the Opt-Miner-Bench in Appendix A.1.

**Baselines** We compare our method against representative baselines categorized into four distinct groups. The commercial LLMs include models such as GPT-4o (OpenAI, 2024) and OpenAI-o3 (OpenAI et al., 2025). The open-source LLM category consists of DeepSeek-V3 (DeepSeek-AI et al., 2025), DeepSeek-R1 (DeepSeek-AI, 2025), and Qwen3 (Yang et al., 2025) models. We evaluate prompt-

based modeling systems, including CoE (Xiao et al., 2024), OptiMUS (Ahmaditeshnizi et al., 2024), and AutoFormulation (Astorga et al., 2025), alongside specialized fine-tuned agents such as ORLM (Huang et al., 2025), Step-Opt (Wu et al., 2025), LLMOPT (Jiang et al., 2025), OptMATH (Lu et al., 2025), and SIRL (Chen et al., 2026), which are predominantly built upon LLaMA or Qwen architectures.

**Implementations** Opt-Miner is implemented using Qwen3-4B-Instruct-2507 and Qwen3-8B as backbones and is equipped with a web research tool of Arxiv Search API to retrieve the top-10 papers for autonomous information seeking, and MinerU API (Niu et al., 2025; Wang et al., 2024) for PDF file transfer. We also use a Qwen3-32B model for file summary. The agent is trained on 2k data with 1k OptMATH data, 0.5k Opt-Miner data generated with the union operator, and 0.5k data with the transfer operator. All the synthesis data are applied with the knowledge fogging operator. We use DeepSeek-V3 for data synthesis.

**Metric** Following existing works in LLM optimization modeling (Ahmaditeshnizi et al., 2024; Lu et al., 2025), we use modeling accuracy as the metric. A model is considered accurate only if it yields an optimal objective value equal to the ground-truth value executed by the Gurobi solver.

## 5.2. Performance of Opt-Miner Agent

As illustrated in Table 1, Opt-Miner achieves high performance across all seven mathematical modeling benchmarks. Opt-Miner-Qwen3-8B achieves performance comparable

*Table 2.* Ablation study on training results across different data in Opt-Miner. We evaluate the impact of different training data on modeling accuracy (Pass@1). We use the Qwen3-4B-Instruct model as the backbone.

| Data | NL4Opt | MAMO | | IndustryOR | Resocratic | OptMATH | Opt-Miner-Bench |
|---|---|---|---|---|---|---|---|
| | | EasyLP | ComplexLP | | | | |
| OptMATH-1k | 88.9 | 86.6 | 52.1 | 28.0 | 54.4 | 27.1 | 7.0 |
| OptMATH-2k | 89.6 | 90.1 | 54.5 | 32.0 | 56.7 | 28.3 | 11.7 |
| OptMATH-1k + Union 1k | 92.3 | 90.4 | 59.2 | 39.0 | 56.9 | 34.3 | 17.2 |
| OptMATH-1k + Transfer-1k | 93.7 | 91.3 | 62.6 | 36.0 | 58.9 | 33.7 | 18.0 |
| Opt-Miner w/o Fogging | 89.9 | 90.0 | 56.3 | 34.0 | 58.5 | 30.7 | 14.8 |
| Opt-Miner Full | 96.2 | 92.5 | 66.4 | 41.0 | 60.6 | 35.5 | 21.1 |

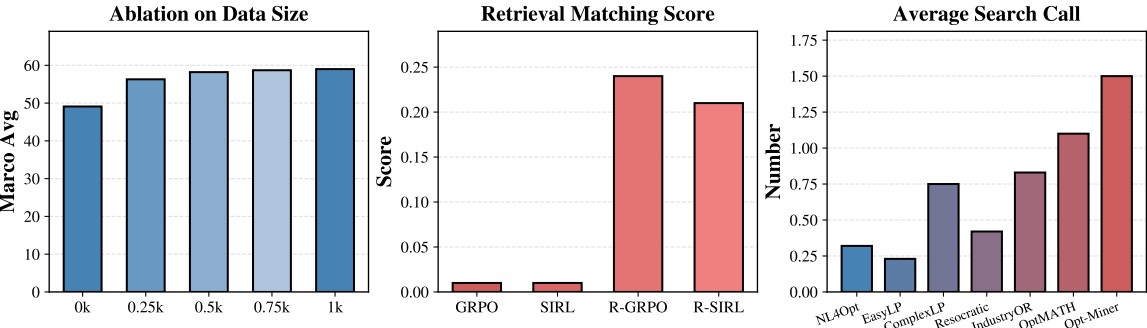

*Figure 3.* **Left**: Ablation on the training data size. The data size 0.25k means we use 1k OptMATH data and 0.25k Opt-Miner training data. **Middle**: The comparison of retrieval quality for different RL algorithms. **Right**: The average search call number of Opt-Miner.

to 32B state-of-the-art specialized agents and commercial reasoning models. Key observations include three aspects. First, it delivers superior modeling accuracy. Even with a relatively small 4B backbone, our method surpasses larger fine-tuned modeling agents, and our fine-tuned Qwen3-8B model achieves comparable performance to 32B models. Second, it bridges the knowledge gap. On more challenging and knowledge-intensive benchmarks such as IndustryOR, OptMATH and Opt-Miner-Bench, Opt-Miner gains significant advantages over existing fine-tuned models. Third, our method can significantly improve the modeling performance on different backbone models. Both Opt-Miner-Qwen3-4B-Instruct and Opt-Miner-Qwen3-8B have shown superior performance compared to their backbone models.

### 5.3. Ablation Studies on the Training Data

**Analysis on the Synthesis Operator** For Opt-Miner, we construct a 2k training dataset: it consists of 1k OptMATH samples, plus 0.5k synthetic samples generated via the union operator and another 0.5k via the transfer operator. To assess the contribution of each module in our data synthesis pipeline, we performed a systematic ablation study as summarized in Table 2. Taking OptMATH-1k and OptMATH-2k as the baseline, we first observed performance improvements after incorporating the union and transfer operators. Both the OptMATH-1k+Opt-Miner (Union 1k) and the OptMATH-1k+Opt-Miner (Transfer 1k) outperform OptMATH-2k. The final performance of Opt-

Miner further surpasses that of OptMATH-2k, confirming that expanding domain diversity and structural complexity via sub-tree evolution effectively boosts the model's capacity to tackle multi-dimensional industrial optimization problems. Notably, the knowledge fogging operator emerges as a critical component for lifting the agent's final performance. When this operator is excluded (Opt-Miner w/o Fogging), the model's performance drops noticeably compared to the full Opt-Miner setup.

**Analysis on the Data Size** We investigate the impact of training data scale on model performance, with results presented in the left panel of Figure 3. As the data size expands initially, the macro-average accuracy across benchmarks increases notably——this reflects the model's rapid capture of core modeling patterns with more training data in the early training stage. Once the data scale reaches a moderate level, the model's performance becomes stable.

**Analysis on the Data Complexity** We analyze the distribution and complexity of our benchmark compared to existing benchmarks. As shown in Table 6 in Appendix A, our dataset exhibits broader domain coverage. Second, as shown in Figure 4, the synthesized problems demonstrate higher textual complexity with a longer average problem length. Finally, these problems feature a richer internal structure with much more involved entities. By incorporating a larger number of interacting entities and constraint clusters, we construct more complex scenarios.

*Table 3.* Ablation study of the reward function. We evaluate the impact of different reward components on modeling accuracy (Pass@1). We use the Qwen3-4B-Instruct model as the backbone.

| Method | NL4Opt | MAMO | | IndustryOR | Resocratic | OptMATH | Opt-Miner-Bench |
| | | EasyLP | ComplexLP | | | | |
|---|---|---|---|---|---|---|---|
| GRPO | 90.3 | 88.7 | 57.3 | 35.0 | 54.0 | 29.5 | 16.4 |
| SIRL | 95.1 | 91.3 | 57.8 | 36.0 | 57.8 | 30.7 | 18.8 |
| R-GRPO | 96.2 | 92.5 | 66.4 | 41.0 | 60.6 | 35.5 | 21.1 |
| R-SIRL | 96.5 | 91.9 | 64.0 | 43.0 | 62.8 | 34.3 | 19.5 |

*Table 4.* Ablation study of the web research module. We evaluate the impact of different reward components on modeling accuracy (Pass@1). We use the Qwen3-4B-Instruct model as the backbone.

| Method | NL4Opt | MAMO | | IndustryOR | Resocratic | OptMATH | Opt-Miner-Bench |
| | | EasyLP | ComplexLP | | | | |
|---|---|---|---|---|---|---|---|
| Qwen3-4B-Instruct | 74.1 | 65.3 | 15.2 | 19.0 | 40.8 | 22.9 | 5.5 |
| +Web Research | 80.6 | 72.6 | 20.9 | 27.0 | 40.5 | 27.7 | 7.0 |
| DeepSeek-V3 | 79.8 | 95.2 | 53.2 | 36.0 | 55.1 | 23.4 | 23.4 |
| +Web Research | 92.7 | 95.7 | 59.2 | 40.0 | 58.3 | 33.1 | 28.1 |
| Opt-Miner w/o Web Research | 93.8 | 91.9 | 57.8 | 35.0 | 55.7 | 28.3 | 12.5 |
| Opt-Miner | 96.2 | 92.5 | 66.4 | 41.0 | 60.6 | 35.5 | 21.1 |

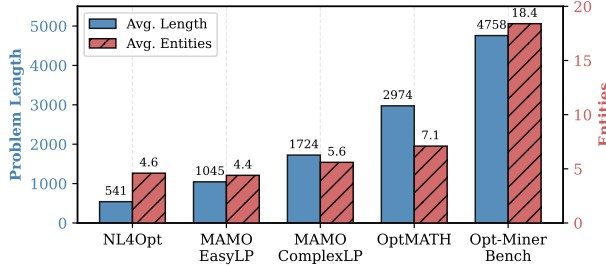

*Figure 4.* Comparison of data complexity. Opt-Miner exhibits significantly higher textual and structural complexity.

*Table 5.* Comparison of average inference time per problem.

| Method | Time (s) | Accuracy (Macro Avg) |
|---|---|---|
| Qwen3-8B | 52.5 | 38.7 |
| ORLM-Llama3-8B | 9.4 | 43.0 |
| OptMATH-Qwen2.5-7B | 13.2 | 49.3 |
| SIRL-Qwen2.5-7B | 35.6 | 53.4 |
| **Opt-Miner-Qwen3-8B** | 47.6 | 65.0 |

## 5.4. Ablation Studies on the Training Algorithms

**Analysis on the Reward Functions**   We conduct experiments to evaluate training algorithms, comparing our proposed R-GRPO on Qwen3-4B-Instruct with three reinforcement learning variants: GRPO (excluding retrieval matching reward), SIRL (from the original paper (Chen et al., 2026) with partial KL term), and R-SIRL (integrating retrieval matching score and partial KL term). Results in Table 3 show that inforamtion seeking significantly improves basic reinforcement learning methods by guiding targeted, efficient information seeking. The retrieval matching reward (keyword matching F1 score) in the middle of Figure 3 confirms it effectively boosts retrieval quality.

## 5.5. Analysis on the Agentic Framework

**Analysis on the Web Research Module**   Table 4 presents the ablation study of the web research module, evaluating its impact on modeling accuracy. Qwen3-4B-Instruct+Web Research represents the base Qwen3-4B-Instruct model equipped with the web research module. Compared to Qwen3-4B-Instruct, Qwen3-4B-Instruct+Web Research achieves improved performance across all benchmarks. Re-

moving the web research module (Opt-Miner w/o Web Research) leads to performance degradation, confirming the module's effectiveness. Notably, the gain from web research is more prominent on challenging benchmarks (e.g., MAMO ComplexLP, IndustryOR, OptMATH, Opt-Miner-Bench), indicating the module better supports the agent in addressing knowledge-intensive, complex problems.

**Analysis on the Tool Use**   We report the average number of search calls across datasets in the right of Figure 3. For simpler datasets such as NL4Opt, MAMO EasyLP, and Resocratic, the agent makes fewer search calls, as these problems can be solved almost entirely using the agent's internal knowledge without relying on external information. For more challenging datasets, including OptMATH and Opt-Miner, the agent encounters significant knowledge gaps, prompting it to increase search calls to acquire the necessary external knowledge for problem-solving.

**Inference Efficiency**   We evaluate the inference efficiency by measuring the average wall-clock time required to solve a single problem on an NVIDIA 80G GPU. As shown in Table 5, while our method introduces slightly more time overhead, Opt-Miner has a stronger performance.

## 6. Conclusion

This work proposes Opt-Miner to address domain knowledge gaps in LLM-based industrial optimization modeling. The framework integrates hierarchical scenario tree-based data synthesis, autonomous web research, and hybrid reward-driven training, enabling the agent to acquire external knowledge and handle complex, underspecified problems proactively. Experiments across seven benchmarks confirm Opt-Miner's superior performance, with notable gains on knowledge-intensive industrial scenarios.

## Acknowlegement

This work was supported in part by the National Key R&D Program of China under contract 2022ZD0119801, National Nature Science Foundations of China grants U23A20388 and 62021001. We would like to thank all the anonymous reviewers for their insightful comments.

## Impact Statement

This paper presents work whose goal is to advance the field of machine learning. There are many potential societal consequences of our work, none of which we feel must be specifically highlighted here.

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

# A. Details of Opt-Miner Datasets

## A.1. Analysis on the Opt-Miner Benchmark

**Diverse Domain Coverage**    As shown in Table 6, Opt-Miner achieves extensive coverage by reformulating complex problems from both academic literature and industrial cases into a hierarchical tree structure. By utilizing sub-tree union and transfer operators, the framework synthesizes multi-domain tasks that decouple mathematical logic from specific industrial scenarios. This approach allows the benchmark to encompass a wide array of physical entities and constraint clusters.

*Table 6.* Scenario Distribution across Datasets (Percentages)

| Scenario | NL4OPT | MAMO Easy | MAMO Complex | IndustryOR | Optibench | OptMATH | Opt-Miner |
|---|---|---|---|---|---|---|---|
| Agriculture | 6.12% | 4.60% | 2.37% | 5.00% | 7.60% | 0.00% | 2.34% |
| Aviation | 0.00% | 0.00% | 0.95% | 2.00% | 0.33% | 7.25% | 3.13% |
| Construction | 0.82% | 6.60% | 0.47% | 1.00% | 1.32% | 0.52% | 0.78% |
| Education | 0.82% | 4.91% | 0.00% | 1.00% | 1.16% | 0.00% | 1.56% |
| Energy | 1.22% | 4.60% | 2.37% | 0.00% | 2.48% | 3.63% | 4.69% |
| Environment | 0.00% | 5.83% | 0.00% | 0.00% | 0.00% | 0.00% | 2.34% |
| Finance | 2.04% | 10.74% | 6.64% | 7.00% | 3.64% | 5.70% | 4.69% |
| Healthcare | 12.65% | 5.06% | 3.79% | 1.00% | 3.31% | 0.00% | 5.47% |
| Logistics | 6.53% | 1.53% | 14.69% | 8.00% | 6.94% | 12.95% | 18.75% |
| Manufacturing | 31.84% | 4.91% | 5.21% | 37.00% | 41.98% | 36.27% | 5.47% |
| Marketing | 1.22% | 6.44% | 0.00% | 0.00% | 1.82% | 0.00% | 3.13% |
| Military | 0.00% | 5.67% | 0.00% | 2.00% | 0.50% | 0.52% | 2.34% |
| Retail | 4.08% | 5.06% | 3.32% | 4.00% | 3.14% | 0.00% | 1.56% |
| Services | 10.20% | 10.89% | 2.37% | 10.00% | 6.61% | 4.15% | 3.13% |
| Sports | 0.00% | 4.75% | 0.00% | 0.00% | 0.00% | 0.00% | 0.78% |
| Supply Chain | 0.82% | 5.06% | 27.01% | 7.00% | 1.82% | 7.77% | 6.25% |
| Technology | 0.00% | 0.92% | 4.27% | 1.00% | 0.50% | 0.52% | 12.50% |
| Telecom. | 0.00% | 3.68% | 1.42% | 0.00% | 0.17% | 10.88% | 7.03% |
| Transportation | 16.73% | 7.67% | 19.43% | 8.00% | 6.61% | 5.18% | 7.03% |
| Other | 4.90% | 1.07% | 5.69% | 6.00% | 10.08% | 4.66% | 7.03% |

**Diverse Type Coverage**    As shown in Table 7, Opt-Miner has selected a diverse set of 128 representative mathematical optimization problems spanning multiple optimization types, including IP, LP, NLP, and SOCP, with a distinct emphasis on MILP. This focus allows the benchmark to capture the complexity of realistic industrial scenarios, enabling extensive coverage of multi-domain tasks across various sectors.

**Preventing Data Leakage**    Seed problems were extracted from premier Operations Research journals (e.g., OR, MS, EJOR) and adapted from real-world supply chain cases. Selection was based on three criteria: broad domain coverage, problem diversity, and high complexity (requiring at least 10 categories of constraints). We applied a difficulty filter using GPT-4o and DeepSeek-V3 to retain only instances where these models achieved less than 50% accuracy across five attempts. All synthesized problems underwent manual inspection by domain experts to ensure semantic coherence and the correctness of the Gurobi solver code. Crucially, the use of Scenario Union and Scenario Transfer ensures that the synthesized problems are entirely new, thereby eliminating the risk of data leakage where an agent might otherwise retrieve original solutions from the web.

*Table 7.* Optimization Type Distribution across Datasets (Percentages)

| Scenario | NL4OPT | MAMO Easy | MAMO Complex | IndustryOR | Optibench | OptMATH | Opt-Miner |
|---|---|---|---|---|---|---|---|
| IP | 26.94% | 58.74% | 15.64% | 35.00% | 15.06% | 11.92% | 2.34% |
| LP | 61.63% | 0.61% | 62.08% | 32.00% | 9.64% | 8.81% | 5.47% |
| MILP | 11.43% | 40.65% | 22.28% | 30.00% | 55.42% | 67.87% | 82.03% |
| NLP | 0.00% | 0.00% | 0.00% | 3.00% | 0.00% | 5.18% | 7.03% |
| SOCP | 0.00% | 0.00% | 0.00% | 0.00% | 19.88% | 6.22% | 3.13% |

# B. Formal Definition of Hierarchical Scenario Tree

To clarify the mechanism by which Opt-Miner scales problem complexity, we provide a formal definition of the Hierarchical Scenario Tree (HST). An HST is represented by the tuple $T = \langle \mathcal{R}, \mathcal{V}_E, \mathcal{V}_C, \mathcal{Q} \rangle$. Here, $\mathcal{R}$ denotes the root node containing the global scenario context $S$. $\mathcal{V}_E$ is the set of nodes in the entity subtree $T_E$, where each node $e \in \mathcal{V}_E$ is defined by functional attributes $\mathcal{A}(e)$. $\mathcal{V}_C$ represents the set of nodes in the constraint subtree $T_C$, where each node $c \in \mathcal{V}_C$ represents

a mathematical constraint and its natural language description $D_c$. Finally, $\mathcal{Q}$ is the ground-truth retrieval evidence set recorded during synthesis. The HST representation possesses two key properties: Structural Decoupling, which allows the constraint logic to be transplanted onto different entity sets via an injective mapping $\Phi : \mathcal{V}_{Ei} \rightarrow \mathcal{V}_{Ej}$, and the Union Property, where two compatible trees ($S_{sem} = 1$) can be merged into a valid $T_{syn}$ without violating internal consistency.

## C. Structural Correspondence Evaluation

While matching the optimal objective value is the primary performance metric, we also assess the structural alignment between agent-generated models and ground truths. This multi-dimensional evaluation utilizes key statistical features as proxy indicators of modeling proficiency. We report the proportion of models that match the ground truth across variable counts, binary/integer variable counts, and the number of constraints. As shown in Table 8, Opt-Miner achieves the highest rate of structural correspondence on the OptMATH dataset. This suggests that the agent not only identifies the correct numerical answer but also internalizes the underlying mathematical structure. It is important to note that a discrepancy in these counts does not inherently mean a model is invalid, as optimization problems can often be correctly formulated in several mathematically equivalent ways.

*Table 8.* Structural Alignment Accuracy on the OptMATH Dataset (%)

| Method | Var Match | Bin Match | Cons Match |
|---|---|---|---|
| DeepSeek-V3 | 39.7 | 44.6 | 28.9 |
| GPT-4o | 27.1 | 33.1 | 24.6 |
| ORLM (8B) | 15.1 | 12.7 | 9.0 |
| OptMATH (32B) | 45.2 | 41.5 | 31.9 |
| SIRL (32B) | 56.0 | 48.7 | 50.0 |
| **Ours (8B)** | **57.8** | **53.0** | **53.6** |

# D. Training and Inference Details

## D.1. Training Setup and Hyperparameters

All training experiments for the Opt-Miner framework covering Qwen3-4B-Instruct and Qwen3-8B variants were conducted on a single computing node with eight NVIDIA 80G GPUs. The proposed Retrieval-based Group R-GRPO was implemented with the VeRL framework. The agent was trained under a multi-turn trajectory format that integrates reasoning thoughts, autonomous search queries and external document observations. Detailed hyperparameters for the R-GRPO training phase are summarized in Table 9.

*Table 9.* List of training hyperparameters for Opt-Miner (R-GRPO).

| Category | Parameter | Value |
|---|---|---|
| Data | Training Epochs | 5 |
| | Training Batch Size | 128 |
| | Max Sequence Length | 8192 |
| | Max Prompt Length | 3300 |
| | Learning Rate | $1 \times 10^{-6}$ |
| | Optimizer | AdamW |
| | Warmup Ratio | 0.05 |
| R-GRPO | Group Size | 8 |
| | KL Coefficient | 0.01 |
| | Clip Range | 0.2 |
| Reward | Accuracy Weight ($\omega_a$) | 1.0 |
| | Retrieval Weight ($\omega_r$) | 0.3 |

## D.2. Inference Configuration and Research Loop

During the evaluation stage, temperature was set to zero to ensure the reproducibility of mathematical modeling results and autonomous research trajectories. The Opt-Miner agent was configured to conduct up to three turns of iterative web research for each problem instance. In each turn, if the agent generated a specific search tag `<search>...</search>`, the generation process was temporarily suspended to invoke the Arxiv Search API. Retrieved technical documents were processed via the MinerU API to extract clean text from PDF files and summarized by a Qwen3-32B model, with the resulting summaries then fed back into the context window within an observation tag `<result>...</result>`. Once the agent judged that sufficient information had been gathered or the maximum turn limit had been reached, it generated the definitive mathematical formulation and corresponding executable Gurobi code within tags `<python>...</python>`. All generated code was executed and debugged in a standardized Python 3.10 environment to verify objective values against the ground truth. Specific parameters for inference are detailed in Table 10.

## D.3. Implementation Details for Inference Efficiency

An important thing is to consider the efficiency-efficiency trade-off (Mao et al., 2026). The iterative information-seeking process of Opt-Miner involves multiple rounds of web searching and document processing. To mitigate potential latency issues, we implemented several engineering optimizations. First, a caching mechanism stores results for identical search queries and previously accessed URLs. Second, we utilize parallel document retrieval to visit and process multiple web pages simultaneously. Furthermore, we established strict execution constraints: a maximum timeout of 15 seconds per network request and a limit of 3 retry attempts for failed links. These measures ensure that the total inference time remains acceptable. As shown in Table 11, while web retrieval adds a search overhead, the grounded reasoning reduces the overall LLM generation time compared to base models by minimizing repetitive reasoning and hallucinations.

*Table 10.* Inference and environment settings for Opt-Miner.

| Parameter | Value |
|---|---|
| Decoding Method | Greedy Search |
| Temperature | 0.0 |
| Top-p | 0.8 |
| Top-k | 20 |
| Max Research Turns | 3 |
| Backbone Solvers | Gurobi |

*Table 11.* Network and Inference Statistics for Opt-Miner Agent

| Metric | Value |
|---|---|
| Link Fetching Success Rate | 93.3% |
| Average Retry Count per Query | 0.14 |
| Average Search Time per Problem | 20.8s |
| Average LLM Generation Time per Problem | 25.8s |

## E. Comparison with SFT Baselines

To evaluate the advantages of the reinforcement learning approach employed in Opt-Miner, we conducted a comparative analysis between our Retrieval-based Group Relative Policy Optimization (R-GRPO) and a standard Supervised Fine-Tuning (SFT) baseline. The comparative results across different benchmarks are organized in Table 12, which illustrates the performance gap between the static imitation of SFT and the dynamic, reward-driven exploration of Opt-Miner.

*Table 12.* Performance comparison between SFT baseline and Opt-Miner across benchmarks.

| Method | NL4Opt | MAMO | | IndustryOR | Resocratic | OptMATH | Opt-Miner-Bench |
|---|---|---|---|---|---|---|---|
| | | EasyLP | ComplexLP | | | | |
| SFT Baseline | 87.5 | 84.9 | 42.1 | 22.0 | 47.5 | 19.8 | 6.3 |
| Opt-Miner (Ours) | 96.2 | 92.5 | 66.4 | 41.0 | 60.6 | 35.5 | 21.1 |

## F. Error Analysis

We conduct an error analysis to comprehend the Opt-Miner framework, particularly in scenarios where the agent failed to reach the ground-truth optimal value. We categorized the observed failures into three primary modes: retrieval inefficiency, execution failure, and logical modeling error. Retrieval inefficiency occurs when the agent generates misaligned search queries, resulting in irrelevant technical documentation that adds noise to the context window. Execution failure refers to cases where the solver code cannot run successfully. Lastly, modeling errors involve the cases have correct retrieved information and successful code execution but wrong answer, such as misdefining variables or constraints.

## G. Further Experiments

### G.1. Reward Weight Analysis

We analyze the impact of varying the retrieval matching reward weight $\omega_r$ on performance. As $\omega_r$ increases from a low value, the model's performance rises notably—this is because moderate weight effectively guides targeted information seeking. When $\omega_r$ exceeds a threshold, performance begins to decline, as excessive emphasis on retrieval distracts the model from final modeling accuracy. The result shows there exists an optimal $\omega_r$ that balances the supervision of information seeking and mathematical formulation.

*Table 13.* Distribution of error modes for the base model and Opt-Miner (Qwen3-4B-Instruct) in the IndustryOR dataset.

| Error Category | Retrieval Failure | Execution Failure | Modeling Error |
|---|---|---|---|
| Base Model | 48.0 | 32.0 | 49.0 |
| Opt-Miner (Ours) | 16.0 | 19.0 | 40.0 |

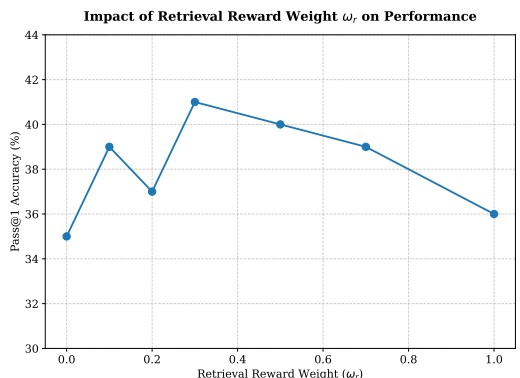

*Figure 5.* The reward sensitivity analysis.

## H. A Simplified Example of HST

**Job Shop Scheduling**

1. **Scenario Domain**: Logistics.

2. **Scenario Description**: A logistics operator manages a fleet of **Heterogeneous Vehicles** (varying in capacity, speed, and cost) to service a set of $N$ geographically distributed customers. Each customer requires a specific service within a strict **Time Window**. The drivers operate under strict **Hours-of-Service (HOS)** regulations, mandating rest breaks after continuous driving. The goal is to design a set of routes that service all customers, respecting vehicle and driver limitations.

3. **Objective**: Minimize Fleet Cost. Primary goal is to minimize the number of vehicles used and the total distance traveled, weighted by vehicle type costs.

4. **Core Entities & Parameters**:

    (a) **Customers** ($C$): Delivery points with demand and time windows $[e_i, l_i]$.
    (b) **Heterogeneous Fleet** ($K$): Vehicles with distinct Payload $Q_k$ and Max Duration $D_k$.
    (c) **HOS Rules**: Mandatory break duration after set driving hours.

5. **Mathematical Constraints**:

    (a) **Vehicle Capacity**: The total load assigned to route $k$ cannot exceed the vehicle's specific payload $Q_k$.
    (b) **Route Duration Limit**: The total time (Travel + Service + Wait) for route $k$ must not exceed $D_k$.
    (c) **Time Windows**: Vehicle arrival at customer $i$ must be within $[e_i, l_i]$. Early arrivals wait; late arrivals are forbidden.
    (d) **Driver Rest Mandate**: If the cumulative driving time since the last rest exceeds $T_{drive}^{max}$, a mandatory break of duration $T_{break}$ must be inserted into the schedule before driving can resume.
    (e) **Site-Vehicle Compatibility**: Certain customers can only be visited by specific vehicle types (e.g., small urban vans).

# I. Example of Generated Problems

## I.1. Union

### Problem 1 to Union

A central warehouse manages the daily vaccine inventory for three community clinics over a three-day planning horizon, distributing sensitive products with a fixed two-day shelf life. On the morning of Day 1, Clinic 1 opens with zero stock and a forecasted demand of 60 doses, necessitating a large shipment that must cover this demand and restore the mandatory 20-unit safety buffer without exceeding the facility's 200-unit storage capacity. Clinic 2 starts with 50 units of one-day-old stock, but with a daily demand of only 30 doses, the surplus 20 units will reach their expiration limit tonight and incur a $50 per unit spoilage penalty. Clinic 3 presents a more complex inventory mix, holding 15 units of one-day-old stock and 10 units of fresh stock against a demand of only 15 doses; while the older stock is consumed first to minimize waste, the remaining 10 units of fresh stock will fall below the safety threshold, requiring a precise non-negative replenishment order that balances the risk of a $100 shortage fine against the $2 per unit overnight holding cost.

### Problem 2 to Union

MetroLogistics initiates its delivery operations on Tuesday at 08:00 AM with the primary objective of **minimizing total fleet scheduling costs**. The fleet consists of one heavy-duty truck (3,000 kg capacity), one electric van (800 kg capacity, strictly limited to a 4-hour maximum endurance), and one cargo bicycle (50 kg capacity). The day's itinerary involves four specific customers: Customer A, located in an industrial zone, requires a delivery of 2,600 kg by 10:00, with a travel time of 20 minutes from the depot; Customer B, situated in a vehicle-restricted pedestrian zone that permits only bicycle access, requires 10 kg by 12:00, with a round-trip duration of 50 minutes; Customer C awaits 600 kg by a strict deadline of 11:30, and the travel time from Customer A to Customer C is 30 minutes; finally, Customer D in the outer suburbs requires 150 kg by 14:00, with a projected total mission time of 3 hours and 50 minutes. Operational regulations mandate that drivers must take a mandatory 45-minute break after reaching 4 hours of continuous driving. At the 08:00 start time, the heavy truck driver has already accumulated 3 hours and 40 minutes of continuous driving, while the other vehicles begin in a fresh state. The scheduling solution must determine the lowest-cost routing plan while strictly adhering to all time windows, capacity limits, access restrictions, and rest break regulations.

### Generated Problem

Scheduling task for the BioLife logistics network involves distributing vaccines to ten regional clinics, labeled C01 through C10, using a fleet of five heterogeneous thermal vans. All vehicles depart from the central warehouse at 08:00. The operation adheres to a Vendor-Managed Inventory (VMI) protocol, which mandates that the ending inventory at any node—calculated as the initial inventory minus daily demand plus the new delivery quantity—must not fall below the specific safety stock threshold assigned to that site. The fleet consists of two distinct vehicle types: two Type A vans (V-A1, V-A2) characterized by a maximum payload of 3000 units and a battery range of 480 minutes (8 hours), suitable for high-volume or long-distance routes; and three Type B vans (V-B1 through V-B3) with a maximum payload of 1000 units and a battery range of 330 minutes (5.5 hours). Operational compliance requires that no single vehicle visits more than four clinics per trip to ensure thermal stability, and a mandatory 45-minute rest break must be scheduled if cumulative driving time reaches 240 minutes (4 hours).

Immediate distribution is required for a batch of 2000 vaccine units at the central warehouse that will expire the following day. This batch is assigned to Clinic C08, as it is the only facility with sufficient capacity. Since C08 is located 200 minutes from the warehouse (resulting in a 400-minute round trip), the distance exceeds the battery range of Type B vehicles, and the volume exceeds their payload limit, necessitating the assignment of a Type A vehicle. Clinic C08 operates with an initial inventory of 800, demand of 200, safety stock of 200, a physical storage capacity of 5000, and a reception deadline of 18:00.

Distinct physical and temporal requirements apply to other specific nodes. Clinic C04, located in a narrow urban zone, is physically restricted to Type B vehicles only; it holds an initial inventory of 50 against a demand of 40 and a safety stock of 20, with a capacity of 400 and a reception window closing at 14:00. Clinic C09 requires a travel time of 150 minutes from the warehouse and operates under the network's tightest reception deadline of 13:00, with an inventory state of 120 units, demand of 80, safety stock of 30, and capacity of 500. The remaining clinics possess the following operational

parameters: C01 (Initial 200/Demand 50/Safety 50/Capacity 500/Window 16:00), C02 (180/80/50/600/16:00), C03 (150/60/40/500/15:00), C05 (300/100/80/1000/17:00), C06 (250/50/50/800/17:00), C07 (400/100/100/1200/18:00), and C10 (200/50/50/600/16:00).

Travel time calculations rely on fixed durations: the route from the warehouse to C04 takes 30 minutes, to C09 takes 150 minutes, and to C08 takes 200 minutes. Travel between average adjacent nodes is estimated at 20 minutes, while cross-zone travel from urban to remote sectors requires 100 minutes. The optimization model must utilize these parameters, along with the vehicle payload, battery limits, and driver rest requirements, to determine a delivery plan that satisfies all inventory safety levels.

## I.2. Transfer

**Source Problem**

A precision workshop is tasked with manufacturing three custom parts—a gear, a casing, and a shaft—using a lathe, a milling machine, and a grinder. Each part follows a specific production sequence: the gear requires 2 hours on the lathe followed by 3 hours on the milling machine; the casing needs 4 hours on the milling machine followed by 2 hours on the grinder; and the shaft begins with 3 hours on the grinder before moving to the lathe for 2 hours. Since each machine can only process one part at a time and operations cannot be interrupted once started, parts requiring the same machine must be queued. The objective is to schedule these operations to minimize the total time required to complete all three parts.

**Target Problem**

A hospital ward assigns one certified Charge Nurse and one Staff Nurse to jointly manage Patient A, who is in infectious isolation, and Patient B. Patient A requires a 45-minute lumbar puncture procedure that must be performed specifically by the Charge Nurse; upon completion, the collected sample must be delivered to the lab within 15 minutes, while the patient simultaneously enters a mandatory 60-minute recovery period preventing further interaction. Patient B requires a 10-minute antibiotic administration strictly between 09:00 and 09:30. Transitioning from the infectious Patient A to Patient B necessitates a 20-minute decontamination process, whereas standard turnovers require only 2 minutes. The objective is to schedule these operations to minimize the final completion time of all nursing duties.

**Generated Problem**

In a high-intensity clinical operational day, the scheduling system must coordinate the precise treatment pathways for five participants (P-1 through P-5) to minimize the total cohort completion time, subject to strict resource and safety constraints where all interventions must run continuously once started. Participant P-1 follows a multi-stage sequence beginning with 120 minutes of Leukapheresis on Station L1, followed by 90 minutes of Chemotherapy Pre-conditioning on Station B2, which imposes a mandatory 30-minute stationary observation period during which B2 remains blocked, and concluding with a 45-minute Vector Infusion in Isolation Room R7. This final infusion for P-1 must commence within 60 minutes of the chemotherapy step's completion to ensure drug stability and requires the continuous presence of the ACLS Specialist Team. Participant P-2 requires a 60-minute Plasma Exchange on Station L1; if this session is scheduled after P-1, Station L1 requires a 45-minute deep terminal cleaning due to P-1's infection status, whereas the reverse sequence requires only a 15-minute setup. Participant P-3 undergoes a two-step process on Station E3, starting with a 20-minute Anesthesia Assessment and immediately followed by a 50-minute Tumor Biopsy, which also requires the ACLS Specialist Team and must be strictly timed to complete between 14:00 and 16:00 to align with external pathology logistics. Participant P-4 requires a 180-minute Drug Infusion on Station B2, creating a conflict with P-1; switching Station B2 from P-1 to P-4 requires a 30-minute decontamination setup, while the reverse transition requires only 10 minutes. Finally, Participant P-5 requires a 15-minute Pre-check on Station E3 followed by a 30-minute Reaction Test in Isolation Room R7. A critical system-wide constraint is that the ACLS Specialist Team has a maximum capacity to support only one high-risk task at a time, meaning P-1's final infusion and P-3's biopsy cannot occur simultaneously.

## I.3. Fogging

---

**Source Porblem**

In a ring-shaped power grid composed of three stations, the **user electricity demand** at Station 3 is set at 100 MW for the first hour and jumps to 300 MW in the second hour, where the operation is fundamentally based on satisfying **rules ensuring electricity input equals output** at any given moment. Unit G1 at Station 1 has a generation cost of $20/MWh, with its operating state limited by an **adjustment speed** of 50 MW/h and a **minimum continuous running rule** of 4 hours; conversely, Unit G2 at Station 2 has a generation cost of $90/MWh and possesses a **fast adjustment capability** of 200 MW/h. **According to physical laws, two-thirds of the power from Station 1 flows through the direct connection line, while one-third of the power from Station 2 flows through this line, and this line has a physical upper limit of 150 MW.** Furthermore, the optimization model is required to reserve 30 MW of capacity to meet **backup power requirements**, verified to ensure that this reserve capacity possesses network deliverability across the network. The final scheduling solution space is further restricted to pass **a "what-if" safety test to ensure the system remains stable even if any single line suddenly breaks**, ensuring system stability under single-component contingencies.

---

**Fogged Problem**

In a ring-shaped power grid composed of three stations, the **nodal load (fogged)** at Station 3 is set at 100 MW for the first hour and jumps to 300 MW in the second hour, where the operation is fundamentally based on satisfying **nodal power balance (fogged)** equations at any given moment. Unit G1 at Station 1 has a generation cost of $20/MWh, with its operating state limited by a **ramp rate constraint (fogged)** of 50 MW/h and a **minimum run time constraint (fogged)** of 4 hours; conversely, Unit G2 at Station 2 has a generation cost of $90/MWh and possesses a **high ramp rate (fogged)** of 200 MW/h. **Based on DC power flow constraints, the power transfer distribution factor from Station 1 to the critical line is 0.67, while the sensitivity from Station 2 is 0.33, with the line subject to a thermal stability limit of 150 MW (fogged)**. Furthermore, the optimization model is required to reserve 30 MW of capacity to meet **system reserve constraints (fogged)**, verified to ensure that this reserve capacity possesses network deliverability across the network. The final scheduling solution space is further restricted to pass **N-1 security constraints (fogged)**, ensuring system stability under single-component contingencies.

# J. Details of HST Construction Process

## J.1. Construction Prompts for HST

---

**Construction Prompts for HST**

**Role:** You are an expert Industrial Solutions Architect and Operations Research Specialist.
**Task:** Extract structured technical details for **ALL distinct optimization problems** formulated in the provided academic paper content. Focus strictly on the **specific industrial/real-world application scenarios** and their corresponding mathematical formulations.
**Content (Truncated):**
Title: `{title}`
Body: `{content}`
**Requirements:**

1. **Identify Multiple Models (CRITICAL)**:
   - Papers often present a progression of models (e.g., "Security Constrained Unit Commitment", "Multi-Depot Vehicle Routing Problem", "Nurse Rostering Problem").
   - You MUST extract **EACH** distinct mathematical formulation as a separate entry in the output JSON.
   - **DO NOT** use generic mathematical classifications as the model name (e.g., "MILP", "Mixed-Integer Linear Programming", "Convex Optimization", "Heuristic Algorithm").
   - Key: Use the specific problem name (e.g., "Security Constrained Unit Commitment", "Nurse Rostering Problem").

2. **Extraction Details (For EACH model)**:
   - **Scenario**: A detailed narrative of the industrial background, operational process, and core conflicts specific to this model variant.
   - **Objective**: The specific goal function with LaTeX formulas.
   - **Constraints**: Group specific operational restrictions into logical categories (e.g., "System-Wide Constraints", "Physical Constraints"), for every constraint, A list of specific operational restrictions (e.g., "Generator ramping limits", "Transmission line thermal limits") Use LaTeX.

3. **Entity Analysis & Feature Extraction**:
   - Identify key physical or abstract **Entities** (e.g., "Thermal Generator", "Transmission Line").
   - For each entity, extract:
     - **Parameters**: A list of associated input data/attributes found in formulas (Description + Symbol).
     - **Features**: Analyze the entity's physical nature and operational role within the problem context to explicitly infer its core intrinsic features (e.g., discreteness, spatial properties, or dynamic behavior).

4. **Output Format**: Return a valid JSON object strictly. The root must be a dictionary where keys are the Problem Names.

**Example Output Structure:**
```
{{
  "Problem_Name_1 (e.g. Deterministic SCUC)": {{
    "scenario": "Detailed description...",
    "objective": "Minimize ... $Formula$",
    "constraints": {{
      "System Constraints": [ {{ "Power Balance": "$Formula$" }},{{"Spinning Reserve
  Requirement": "$Formula$" }} ]
    }},
    "entities": {{
      "Entity_Name_1": {{
        "parameter": ["$P_{{g,t}}$ (Power Output)"],
        "features": {{
```

---

```
            "Physical_Existential": {{ "is_discrete": true, "material_state": "
    Solid_Object", "is_consumable": false }},
            "Spatial_Geometric": {{ ... }},
            "Interaction_Topology": {{ ... }},
            "Flow_Conservation": {{ ... }},
            "Temporal_Dynamic": {{ ... }},
            "Stochastic_Objective": {{ ... }}
          }}
        }}
      }}
    }}
  }},
  "Problem_Name_2 (e.g. Robust SCUC)": {{
      "scenario": "This variant considers wind uncertainty...",
      "objective": "Minimize ...",
      "constraints": {{ ... }},
      "entities": {{ ... }}
  }}
}}
```

If no specific application scenario is found, return {}.

## J.2. Similarity Evaluation for two HSTs

---

**Similarity Evaluation for two HSTs**

You are an expert in Operations Research and Domain Modeling.

Your task is to compute the Problem Thematic Compatible between two **Structured Problem Definitions**.

**DEFINITION OF SIMILARITY**

The inputs are structured descriptions of industrial or optimization problems (containing scenarios, objectives, and entities).

1. Evaluate whether the two problems share a **common industrial domain** (e.g., Logistics, Manufacturing, Supply Chain, Healthcare).

2. Focus primarily on the semantic context provided in the "scenario" and "entities" fields.

3. Ignore differences in variable naming (e.g., "x" vs "q") or specific data values. Focus on the *type* of problem being solved.

**SCORING SCALE**

You must return a strict binary score (**0** or **1**):

- **1 (Compatible):** The problems belong to the **same broad industrial domain**. Even if the specific mathematical problem type differs (e.g., scheduling vs. routing), if they serve the same industry, the score is 1.

- **0 (Incompatible):** The problems belong to **different industrial domains** (e.g., Supply Chain vs. Image Processing).

**INPUT DATA**

**Tree A:** `{Sematic_Tree_A}`

**Tree B:** `{Sematic_Tree_B}`

**OUTPUT REQUIREMENT** Return ONLY a valid JSON object. Do not output markdown code blocks ("'json).

Format:

```
{{
    "similarity": <integer, 0 or 1>,
    "reasoning": "<string, concise explanation of the common or differing domains>"
}}
```

---

## J.3. Prompts for the Operators

---

**union operator**

You are an expert-level specialist in Operations Research, Data Science, and complex systems modeling. Your mission is to synthesize a single, comprehensive "Fusion Optimization Problem" that logically weaves multiple abstract concepts into a realistic, large-scale business scenario.

**INPUT DATA**

1. **PROBLEM ANCHOR**: `{anchor}`

2. **KNOWLEDGE CONCEPTS LIST**: `{concept_names}`

3. **CONCEPTUAL DESCRIPTION DICTIONARIES**: `{concept_descriptions}`

**MISSION**

Construct a unified optimization problem where the provided concepts are not just present, but are **interdependent** and **essential**. The problem must be designed such that it is mathematically and operationally impossible to solve without addressing all knowledge points through a multi-stage solution path.

**MANDATORY GENERATION REQUIREMENTS**

1. **Narrative Integration (Most Important)**:

   - Avoid Mechanical Structures: Strictly do not use headers such as "Entities," "Environment," "Constraints," "Variables," or "Stage 1/2/3."
   - Fluent Business Prose: The problem statement must read like a high-level executive summary or a professional case study. Every rule, physical limitation, and numerical data point must be woven naturally into the paragraphs of the story.
   - Contextual Renaming: Translate academic terms (e.g., "Graph G," "Decision Variables," "Objective Function") into specific business objects (e.g., "The regional logistics network," "Daily scheduling choices," "The primary success metric").

2. **Maximal Coherent Integration**:

   - **Unified Ecosystem**: Map all entities from all provided dictionaries into a single environment.
   - **Coherence Over Quantity**: While you must aim to include the maximum number of concepts, priority must be given to logical coherence. Ensure that every selected knowledge point combines naturally into a realistic scenario appropriate for the `{anchor}`.
   - **Fluid Renaming**: Rename academic terms (e.g., "Graph G", "Trotter Register") into business entities (e.g., "Standard Compliance Protocol", "Data Synchronization Buffer").
   - **Entity Expansion**: You are encouraged to introduce any additional auxiliary entities (e.g., weather conditions, regulatory bodies, cost centers) to make the scenario more realistic, complex, and fluent, as long as the core roles from the dictionary remain the decision backbone.

3. **Logical Interdependency (Mandatory)**:

   - **Essentiality**: Concepts must not be merely "mentioned." They must be the structural pillars of the problem. If a single concept or constraint is ignored, the entire problem must become ill-posed or unsolvable.
   - **Constraint Integrity**: Every mathematical bound (e.g., $\epsilon_T, \tau, 1/KQ$) from the dictionaries must be translated into a functional "hard limit" in the business narrative.

4. **Step-by-Step Solution Path**:

   - **Multi-Stage Structure**: The problem must be complex enough to require a multi-stage application of the optimization concepts.
   - **Necessity Logic**: The "Scene Knowledge" (Entities and Environment) must create the specific challenges that necessitate the use of the "Optimization Knowledge" (Mathematical concepts).

---

5. **Scale, Complexity**:

   - The scenario must be large-scale, naturally leading to a high volume of decision variables.
   - The designed problem scenario must be sufficiently large-scale to naturally lead to a model with a very high number of decision variables and a substantial volume of constraints. What's more, your generated problem must explicitly include a minimum of 8 distinct categories of constraints (e.g., resource limits, flow balance, time windows, logical dependencies, etc.), and each constraint must explicitly involve multiple specific variables to ensure sufficient complexity.

6. **Clarity and Coherence**: The final problem statement must be concise, logical, and coherent, even in its complexity.

7. **Scenario Description**: A detailed, multi-paragraph description of the real-world problem. This scenario must naturally and logically weave all knowledge points together. In your generated problem statement, **you must avoid explicit academic Operations Research terminology**. Do not use terms like "objective function", "decision variables", "constraints", or "formulate a mathematical model". Instead, describe these concepts naturally within the business or operational context.

8. **Objective Function**: A clear, formal statement of the primary objective (e.g., "Minimize total operational costs," "Maximize supply chain resilience").

9. **Specific Data and Operational Rules**: The scenario must be richly populated with specific quantitative data and precise operational rules. You must provide concrete numerical values for costs, capacities, demands, rates, time windows, and limits. Furthermore, clearly describe the specific structure of these limitations in natural language (e.g., "the total weight loaded onto any vehicle cannot exceed its rated capacity of 5,000kg," "if production line A is running product X, line B cannot run product Y simultaneously due to power limitations," "a minimum of 10% of the total monthly production must be allocated to reserve inventory").

**RETURN FORMAT**
Return only the final problem in the following JSON format:
{
"problem": "your generated fusion problem statement"
}

---

**transfer operator**

You are an expert-level specialist in Operations Research, Data Science, and complex systems modeling. Your mission is to synthesize a high-complexity "Cross-Domain Fusion Problem" that applies deep technical logic from a "Problem Anchor" domain to a completely unrelated "Inspiration Domain".

**INPUT DATA**

1. **PROBLEM ANCHOR**: `{anchor}` (The core logical domain)

2. **ANCHOR CONCEPTS**: `{concept_descriptions}` (Technical concepts related to the anchor)

3. **CROSS-DOMAIN INSPIRATION**: `{inspiration_description}` (The unrelated domain that provides the scenario setting)

**MISSION** Design a unified optimization problem where the setting, characters, and entities are primarily derived from the **CROSS-DOMAIN INSPIRATION**, but the underlying operational rules and constraints are dictated by the **ANCHOR CONCEPTS**.

**MANDATORY GENERATION REQUIREMENTS**

1. **Cross-Domain Entity Dominance**:

   - **Primary Setting**: You must use the entities from the "Inspiration Domain" as the main characters, assets, and environment of the problem.
   - **Logical Mapping**: Map the technical roles of the "Anchor Concepts" onto these inspiration entities. For example, if the Anchor is "Quantum Computing" and the Inspiration is "Ancient Maritime Trade", a "Qubit" might become a "Specialized Silk Cargo," and a "Gate Operation" might become a "Customs Inspection Process."
   - **Coherence First**: While prioritizing cross-domain entities, ensure the resulting narrative is semantically and logically sound. The fusion must feel like a "real" (though perhaps exotic) business scenario.
   - **Entity Expansion**: You are encouraged to introduce any additional auxiliary entities (e.g., weather conditions, regulatory bodies, cost centers) to make the scenario more realistic, complex, and fluent, as long as the entities from the "Inspiration Domain" remain the decision backbone.

2. **Maximal Constraint Fusion**:

   - **High-Density Integration**: You should aim to integrate **as many constraints as possible** from both the Anchor Concepts and the Cross-Domain Inspiration. Priority should be given to the most defining technical bounds of the core logic.
   - **Constraint "Cloaking"**: Hide the mathematical rigor within the narrative. For instance, a "Trotter Error Bound" must be translated into a "Shipment Fragility Limit" or "Navigational Precision Tolerance" that uses the exact same numerical logic.
   - **Logic Integrity**: The problem must be unsolvable if the core technical constraints from the Anchor Domain are bypassed.

3. **Interwoven Step-by-Step Path**:

   - The problem must require a multi-stage solution where the results of a "Cross-Domain Stage" (e.g., resource allocation in the inspiration setting) are constrained by the "Technical Logic" (e.g., algorithmic scaling rules).

4. **Scale, Complexity**:

   - The scenario must be large-scale, naturally leading to a high volume of decision variables.
   - The designed problem scenario must be sufficiently large-scale to naturally lead to a model with a very high number of decision variables and a substantial volume of constraints. What's more, your generated problem must explicitly include a minimum of 6 distinct categories of constraints (e.g., resource limits, flow balance, time windows, logical dependencies, etc.), and each constraint must explicitly involve multiple specific variables to ensure sufficient complexity.

5. **Clarity and Coherence**: The final problem statement must be concise, logical, and coherent, even in its complexity.

6. **Scenario Description**: A detailed, multi-paragraph description of the real-world problem. This scenario must naturally and logically weave all knowledge points together. In your generated problem statement, **you must avoid explicit academic Operations Research terminology**. Do not use terms like "objective function", "decision variables", "constraints", or "formulate a mathematical model". Instead, describe these concepts naturally within the business or operational context.

7. **Objective Function**: A clear, formal statement of the primary objective (e.g., "Minimize total operational costs," "Maximize supply chain resilience").

8. **Specific Data and Operational Rules**: The scenario must be richly populated with specific quantitative data and precise operational rules. You must provide concrete numerical values for costs, capacities, demands, rates, time windows, and limits. Furthermore, clearly describe the specific structure of these limitations in natural language (e.g., "the total weight loaded onto any vehicle cannot exceed its rated capacity of 5,000kg," "if production line A is running product X, line B cannot run product Y simultaneously due to power limitations," "a minimum of 10% of the total monthly production must be allocated to reserve inventory").

9. **Narrative Integration (Most Important)**:

   - Avoid Mechanical Structures: Strictly do not use headers such as "Entities," "Environment," "Constraints," "Variables," or "Stage 1/2/3."
   - Fluent Business Prose: The problem statement must read like a high-level executive summary or a professional case study. Every rule, physical limitation, and numerical data point must be woven naturally into the paragraphs of the story.
   - Contextual Renaming: Translate academic terms (e.g., "Graph G," "Decision Variables," "Objective Function") into specific business objects (e.g., "The regional logistics network," "Daily scheduling choices," "The primary success metric").

Return only the final problem in the following JSON format:
{
"problem": "your generated cross-domain fusion problem statement"
}

# K. Prompts for Opt-Miner

---

**Prompts for Opt-Miner**

{"role": "system", "content": "You are an Operations Research expert agent to solve complex optimization problems. Given a question, you need to first think. If needed, you can stop the think process and invoke the arxiv search tool to search for the relevant optimization model formulation (e.g., Set Cover Problem, TSP formulation (The query must be a name of optimization problem, not algorithm or scenarios or others)), and python interpreter tool to solve the problem.  If the code raises error in the results, you should continuous debug to ensure it run succefully.  For example, a trajectory: <|think|> think process </|think|> <search> search query </search> <result> search result </result> <|think|> think process </|think|> <python> python code </python> <result> python code results </result> <|think|> code debug or analysis </|think|> <python> python code </python> <result> python code results </result>. \n\nFor the Gurobi python code, here is a template.\n<python>\nimport gurobipy as gp\nfrom gurobipy import GRB\nimport numpy as np\n\n# 1. Create model\nmodel = gp.Model(\"SimpleMIP\")\n\n# 2. Define parameters\nn = 5  # Assume there are 5 variables\n\n# 3. Create multiple variables at once\n# Method 1: Add variables in a loop\n# vtype: GRB.CONTINUOUS for continuous variables, GRB.INTEGER for integer variables, GRB.BINARY for 0-1 variables\nvariables = []\nfor i in range(n):\n    x = model.addVar(lb=0, vtype=GRB.CONTINUOUS, name=f\"x{i}\")\n    variables.append(x)\n\n# Method 2: Create multiple variables using addVars\nvariables = model.addVars(n, lb=0, vtype=GRB.CONTINUOUS, name=\"x\")\n\n# 4. Create multiple constraints at once\n# Method 1: Add linking constraints using a loop\nconstraints = []\nfor i in range(n-1):\n    # Add constraint: x_i + x_{i+1} <= 10\n    cons = model.addConstr(variables[i] + variables[i+1] <= 10, name=f\"linking_{i}\")\n    constraints.append(cons)\n\n# Method 2: Add different types of constraints\n# Resource constraint: sum of all variables <= 20\nmodel.addConstr(gp.quicksum(variables[i] for i in range(n)) <= 20, name=\"resource_limit\")\n\n# Demand constraint: each variable >= 1\nfor i in range(n):\n    model.addConstr(variables[i] >= 1, name=f\"demand_{i}\")\n\n# 5. Set objective function\n# Gurobi supports linear, quadratic, and some nonlinear objectives\nmodel.setObjective(gp.quicksum(3 * variables[i] for i in range(n)), GRB.MAXIMIZE)\n\n# 6. Solve the optimization problem\nmodel.optimize()</python>"},

{"role": "user", "content": "{problem}\n\n Note:\n- Please keep the think process brief.\n - You must invoke python tool finally. The search query <search>...</search> and python code <python>...</python> should be put outside of the think chunk <|think|>...</|think|>.\n- The code must include <python>import gurobipy as gp\nfrom gurobipy import GRB\nimport numpy as np\n\nmodel = gp.Model(\"SimpleMIP\")</python>. Avoid using \"<\" and \">\" in Gurobi constraints; instead, use \"<=\" or \">=\" as appropriate.\n - Carefully determine whether the variable is an integer or a continuous variable.\n - If you find the code cannot execute successfully with errors, please try to correct and debug the code. /no_think"}

---

# L. Examples

<search>Resource allocation problem</search>

```
Arxiv_id 2512.14925: Related optimization models
{'Multiscale Aggregated Hierarchical Attention (MAHA) - Convex Optimization Aggregation
    ': {'scenario': 'A novel architectural framework for Large Language Models (LLMs)
    designed to overcome the quadratic computational bottleneck of standard Multi-Head
    Self-Attention (MHSA). It addresses the challenge of efficiently modeling long-
    context sequences by decomposing the input into multiple hierarchical scales (e.g.,
     fine, medium, coarse granularity) to capture both local syntactic patterns and
    global semantic dependencies. The core industrial application is in scaling
    transformer-based models for tasks requiring long sequences, such as document
    understanding, genomics, and multimodal learning. The specific variant uses a
    convex optimization layer to optimally fuse the attention outputs from different
    scales, treating it as a resource allocation problem to balance computational
    efficiency with contextual fidelity.', 'objective': 'To find the optimal mixing
    weights $\\mathbf{w}$ that minimize
Arxiv_id 2512.14002: Related optimization models
{'Deadline-Constrained Task Offloading and Resource Allocation Problem (DOAP)': {'
    scenario': 'A Vehicular Edge Computing (VEC) system where vehicles generate
    periodic, computation-intensive real-time tasks (e.g., object detection for
    autonomous driving). These tasks have stringent deadlines (e.g., 10-100ms).
    Vehicles can offload tasks to nearby Roadside Units (RSUs) equipped with 5G modules
     (providing wireless bandwidth) and servers (providing computational resources).
    The core conflict is the joint optimization of three decisions: 1) Service
    Deployment (which RSU hosts the service for each task), 2) Bandwidth Allocation (
    how many Resource Blocks (RBs) to allocate for data offloading), and 3)
    Computational Resource Allocation (how many Computing Units (CUs) to allocate for
    task processing). This must be done under the constraints of limited RSU bandwidth
    and computational capacities, task deadlines, and vehicle-RSU accessibility, aiming
     to maximize a system-wide utility (e.g., total en
Arxiv_id 2512.11667: Related optimization models
{'VR-CG User Association and Communication Resource Allocation (First Stage)': {'
    scenario': "This problem addresses the foundational resource allocation for Virtual
     Reality Cloud Gaming (VR-CG) over a 6G network. The scenario involves multiple
    users equipped with Head-Mounted Displays (HMDs) accessing VR-CG applications from
    cloud-based game engines. Users can connect to multiple Base Stations (BSs)
    simultaneously via Multi-Connectivity to improve robustness. The network must
    allocate Physical Resource Blocks (PRBs) from BSs to users, select an initial video
     resolution and frame rate for each user (based on a game-specific 'quality' or '
    performance' mode preference), and ensure that the end-to-end latency (including
    rendering, propagation, transmission, and processing delays) meets the strict
    Motion-to-Photon requirement for an immersive experience. The core conflict is
    balancing high QoE (driven by resolution/frame rate) against limited wireless
    bandwidth and stringent latency constra
Arxiv_id 2512.08485: Related optimization models
{'Local Greedy Data Poisoning Attack (Baseline)': {'scenario': "A state-of-the-art
    adversarial attack strategy in offline reinforcement learning (RL) security
    research. The attacker aims to degrade the performance of an offline RL agent (e.g
    ., for autonomous driving or industrial robotics) by poisoning its static training
    dataset. This method operates under a 'local' paradigm: it first identifies a
    subset of critical data samples (e.g., those with high Temporal Difference error)
    and then independently optimizes a perturbation for each sample, subject to a
    strict, uniform per-sample budget limit. This approach treats all attacked samples
    equally, regardless of their individual impact potential.", 'objective': 'For each
    targeted sample $i$ in the selected subset $S_{\\text{target}}$, maximize the
    induced Bellman/TD error to disrupt value function learning:\n$$\\max_{\\eta_{i}}
    \\mathcal{L}_{a t k}(\\eta_{i}) \\approx | \\delta (s_{i} + \\eta_{i}^{s}, \\dots)
    |$$', 'constraints': {'Per-Sa
Arxiv_id 2512.05946: Related optimization models
{'Human Resource Allocation Problem (HRAP)': {'scenario': 'A personnel dispatch problem
     in a service or operations context where a pool of officers must be assigned to
```

perform specific tasks across multiple scheduled events. Each officer has varying capabilities (task completion times), events have fixed start times, and officers incur travel (transition) times when moving between event locations. The core conflict is the combinatorial assignment of officers to tasks to minimize the overall completion time of the slowest event, balancing individual officer efficiency with logistical travel overhead.', 'objective': 'Minimize the maximum completion time (makespan) across all events. The completion time for an event is the sum of the execution times of its assigned tasks plus the transition times for officers moving to/from that event. The reward function used in the RL formulation is the negative normalized makespan: $r_t = - \frac{\max_{e} \left(\sum_{t} C_{o, e, t} + \sum_{\text{

Arxiv_id 2512.17918: Related optimization models

{'Quantum Cloud Resource Allocation for IoT': {'scenario': 'A cloud-based Quantum Computing as a Service (QCaaS) platform hosts multiple Quantum Processing Units ( QPUs or QNodes) to serve computational requests (QTasks) from resource-constrained Internet of Things (IoT) applications. QTasks, modeled as quantum circuits, arrive dynamically and must be assigned to available QNodes for execution. The system faces the challenge of efficiently mapping incoming QTasks to QNodes to minimize the average task completion time, considering QNode-specific metrics (e.g., CLOPS, pending queue) and QTask requirements (e.g., circuit depth, qubit count). The resource allocation must be performed online by a broker, making it a dynamic scheduling problem.', 'objective': 'Maximize the cumulative reward, which is defined to minimize the average task completion time (T). The immediate reward for mapping a QTask to a QNode is given by: $R = \begin{cases} \frac{1}{T}, & \text{ if QTask can be executed} \\

Arxiv_id 2511.22224: Related optimization models

{'Single IU/EU Pinching Beamforming Optimization': {'scenario': 'A Pinching-Antenna System (PASS) with a single waveguide and N Pinching Antennas (PAs) serves one Information User (IU) and one Energy User (EU) simultaneously. The PAs can be positioned anywhere along the waveguide to form a pinching beam. The system aims to explore the fundamental trade-off between maximizing the data rate for the IU and maximizing the harvested energy for the EU by optimizing the positions of the PAs along the waveguide.', 'objective': 'Maximize the weighted sum of the effective channel gains to the IU and EU by optimizing the PA positions $\mathbf{x}_{\mathrm{p}} = [x_{\mathrm{p}}^{1},\dots,x_{\mathrm{p}}^{N}]^{T}$.\n$$\n\max_{\mathbf{x}_{\mathrm{p}}} \rho \left| \sum_{n=1}^{N} \frac{e^{j \phi_{\mathrm{I}}^{n}}}{\| \psi_{\mathrm{I}} - \psi_{\mathrm{p}}^{n} \|} \right| + (1 - \rho) \left| \sum_{n=1}^{N} \frac{e^{j \phi_{\mathrm{E}}^{n}}}{\| \psi_{\mathrm{E}} - \psi_{\mat

Arxiv_id 2511.17931: Related optimization models

{'Uplink Carrier Aggregation with Self-Interference Constraint': {'scenario': "In mobile networks, Carrier Aggregation (CA) is used to combine multiple carriers to increase uplink data rates for User Equipment (UE). However, for power-constrained UEs, efficient resource allocation is critical. This involves selecting which Secondary Component Carriers (SCCs) to activate (a discrete decision) and allocating transmit power across the Primary Component Carrier (PCC) and activated SCCs (a continuous decision). A key practical challenge is the self-interference ( SI) caused by transmitter non-linearities. When multiple carriers are amplified by a non-linear power amplifier, harmonic/intermodulation products are generated. If a harmonic of an uplink carrier coincides with the UE's own downlink receive frequency, it causes sensitivity degradation in the downlink receiver. The optimization problem must therefore maximize network throughput while managing this SI, making it a mixed discrete-cont

Arxiv_id 2511.16044: Related optimization models

{'Online Assortment Planning with Inventory Shocks': {'scenario': 'A platform (e.g., retail, cloud computing, volunteer matching) sequentially offers assortments of products/resources to arriving customers over a finite horizon. Each product has a per-unit price, a deterministic usage duration (after which it returns to inventory ), and an initial inventory. The platform faces two types of inventory shocks: 1) Endogenous shocks: When a unit is sold, it becomes unavailable for its usage duration before returning. 2) Exogenous shocks: An adversarial process adds unpredictable, non-negative amounts of units to the inventory at the start of each period. Customer choice models (revealed upon arrival) are adversarial but satisfy

```
    weak substitutability and monotonicity. The platform must decide which feasible
    assortment to offer at each time step to maximize total expected revenue, without
    prior knowledge of future customer arrivals or exogenous shocks.', 'objective': '
    Maximize the total expect
Arxiv_id 2511.02500: Related optimization models
{'Deterministic Quadratic Multiple Knapsack Problem (QMKP)': {'scenario': "A
    combinatorial resource allocation problem where a set of items must be assigned to
    multiple knapsacks (resources) with individual capacity limits. Each item has a
    linear profit and weight. Additionally, if two items are placed in the same
    knapsack, they generate an extra 'pairwise' or quadratic profit. This models real-
    world scenarios like manufacturing (assigning tasks to machines where certain task
    combinations are synergistic), project portfolio selection (where projects in the
    same portfolio have complementary benefits), and logistics (loading cargo into
    containers where specific item pairings save space or increase value). The core
    conflict is maximizing total profit (linear + quadratic) without exceeding any
    knapsack's weight capacity.", 'objective': 'Maximize the sum of linear and
    quadratic profits from all assigned items across all knapsacks.\n$$\\max \\sum_{k
    \\in M} \\left(\\sum_{i \\in I_k} p_i + \\
</result>
```

<|think|> The problem described is a linear programming problem with constraints on resources (paint, glitter, glue) and minimum production requirements. The goal is to maximize profit. This is a classic resource allocation problem. We will formulate it as a linear programming problem and solve it using Gurobi. The variables will be the number of large and small art pieces to produce, which are continuous variables. The constraints will ensure that the resources are not exceeded and the minimum production requirements are met. The objective function will be to maximize profit. The code will be written to solve this problem using Gurobi. </|think|>

```python
<python>
import gurobipy as gp
from gurobipy import GRB

# Create model
model = gp.Model("ArtStoreProfitMaximization")

# Define variables
x = model.addVar(lb=5, vtype=GRB.CONTINUOUS, name="large_art_pieces")  # Large art
    pieces, at least 5
y = model.addVar(lb=5, vtype=GRB.CONTINUOUS, name="small_art_pieces")  # Small art
    pieces, at least 5

# Define objective function (maximize profit)
model.setObjective(30*x + 15*y, GRB.MAXIMIZE)

# Define constraints
# Paint constraint: 4x + 2y <= 100
model.addConstr(4*x + 2*y <= 100, name="paint_constraint")

# Glitter constraint: 3x + 1y <= 50
model.addConstr(3*x + y <= 50, name="glitter_constraint")

# Glue constraint: 5x + 2y <= 70
model.addConstr(5*x + 2*y <= 70, name="glue_constraint")

# Solve the model
model.optimize()

# Output the results
print("Optimal solution:")
print(f"Number of large art pieces: {x.x}")
print(f"Number of small art pieces: {y.x}")
print(f"Maximum profit: ${model.objVal}")
</python> <result>
Optimize a model with 3 rows, 2 columns and 6 nonzeros
```

```
Model fingerprint: 0x44507245
Coefficient statistics:
  Matrix range     [1e+00, 5e+00]
  Objective range  [2e+01, 3e+01]
  Bounds range     [5e+00, 5e+00]
  RHS range        [5e+01, 1e+02]
Presolve removed 1 rows and 0 columns
Presolve time: 0.00s
Presolved: 2 rows, 2 columns, 4 nonzeros

Iteration    Objective       Primal Inf.    Dual Inf.      Time
       0    6.7500000e+02   6.244125e+00   0.000000e+00      0s
       3    4.8750000e+02   0.000000e+00   0.000000e+00      0s

Solved in 3 iterations and 0.00 seconds (0.00 work units)
Optimal objective  4.875000000e+02
</result>
```

<|think|> The code executed successfully and found the optimal solution. The optimal number of large and small art pieces that maximize profit while respecting the resource constraints and minimum production requirements is 10 large art pieces and 10 small art pieces, resulting in a maximum profit of $487.50.

