# OpenReview forum: "Opt-Miner: Empowering Information-Seeking Agent with Tree-Guided Data Synthesis for Optimization Modeling"
_ICML.cc/2026/Conference — ICML 2026 regular_

### Official Review · Reviewer_43Fw · 2026-03-12

**Soundness:** 3
**Presentation:** 3
**Significance:** 2
**Originality:** 2
**Overall Recommendation:** 4
**Confidence:** 3

**Summary:**

The manuscript introduces a technically sound pipeline for enhancing the information-seeking capabilities of optimization agents through structured data synthesis and specialized reinforcement learning. The use of Hierarchical Scenario Trees provides a novel way to decouple mathematical logic from industrial contexts, enabling effective domain transfer and complexity scaling. The R-GRPO algorithm successfully addresses the lack of supervised signals for intermediate retrieval steps by utilizing a hybrid reward function. However, several aspects still require further clarification, including the formal definition of the operations in the Hierarchical Scenario Tree, the evaluation protocol used to assess modeling accuracy, and the clarity of specific mathematical notations in the policy optimization section.

**Compliance With Llm Reviewing Policy:**

Affirmed.

**Final Justification:**

Thanks for your response. My concerns have been solved and I will keep my positive score.

**Key Questions For Authors:**

See Weakness.

**Limitations:**

yes

**Strengths And Weaknesses:**

### Strengths

- The formulation of problems into Hierarchical Scenario Trees allows for the explicit embedding of context and techniques in subtrees, providing a structured foundation for synthesis.

- The "Knowledge Fogging" operator is highly effective at simulating real-world underspecification, forcing the agent to move beyond surface-level translation to deep structural comprehension.

- The framework shows significant gains in "long-tail" industrial scenarios, improving modeling accuracy by over 20% on difficult benchmarks.

### Weaknesses

- While the hierarchical scenario tree is described conceptually, the paper does not provide a formal definition or theoretical properties of the representation.

- The hybrid reward definition combines multiple reward components but does not clearly define the format reward term R_format.

- Modeling accuracy is defined solely based on matching the optimal objective value with ground truth.

---

> ### Author Rebuttal · Authors · 2026-03-30
>
> # Weakness 1
>
> We sincerely thank the reviewer for this constructive suggestion.
>
> - **Formal Definition of HST**. We define a Hierarchical Scenario Tree $\mathcal{T}$ represented by the tuple:
>   $$\mathcal{T} = \langle \mathcal{R}, \mathcal{V}\_E, \mathcal{V}\_C, \mathcal{Q}\rangle$$
>   where:
>   *   **$\mathcal{R}$** is the root node, containing the global scenario context $S$.
>   *   **$\mathcal{V}\_E$** is the set of nodes in the entity subtree $\mathcal{T}_E$. Each node $e \in \mathcal{V}\_E$ is defined by a set of functional attributes $\mathcal{A}(e) = \\{a\_1, a\_2, \dots\\}$.
>   *   **$\mathcal{V}\_C$** is the set of nodes in the constraint subtree $\mathcal{T}\_C$. Each node $c \in \mathcal{V}\_C$ represents a mathematical constraint, containing a natural language description $D\_c$.
>   *   **$\mathcal{Q}$** is the ground-truth retrieval evidence set.
> - **Theoretical Properties**. The HST representation possesses several key properties that enable our data synthesis operators:
>   *   **Property 1 (Structural Decoupling)**. The separation of $\mathcal{T}\_E$ and $\mathcal{T}\_C$ allows for scenario transfer. Formally the condition for scenario transfer is as follows: given two trees $\mathcal{T}\_i$ and $\mathcal{T}\_j$, if there exists an injective mapping function $\Phi: \mathcal{V}\_{E\_{i}} \rightarrow \mathcal{V}\_{E\_{j}}$ such that attributes are preserved, the constraint logic $\mathcal{T}\_{C,i}$ can be orthogonally transplanted onto $\mathcal{V}\_{C,j}$ without violating the internal consistency of the mathematical model.
>   *   **Property 2 (Union Property).** Given two highly compatible trees $\mathcal{T}\_1$ and $\mathcal{T}\_2$ (where, $S\_{sem}(\mathcal{T}\_i, \mathcal{T}\_j) = 1)$, the merged tree $\mathcal{T}\_{syn} = (\mathcal{R}\_{syn}, \mathcal{T}\_{E\_1} \cup \mathcal{T}\_{E\_2}, \mathcal{T}\_{C\_1} \cup \mathcal{T}\_{C\_2})$ remains a valid HST. The union of attribute sets natively resolves entity deduplication.
>
>
> We believe this formalization clarifies the mechanism by which Opt-Miner scales problem complexity and evaluates information-seeking accuracy.
>
> # Weakness 2
>
> Thank you for pointing out this omission. The format reward $R_{format}$ is a simple binary reward designed to ensure the agent correctly uses the required tags (e.g., `<|think|>`, `<search>`, `<python>`) during its reasoning and tool-invocation steps. Specifically, it is defined as:
>
> *   **$R\_{format} = 1$** if the generated trajectory strictly follows the predefined structural format with correctly paired and ordered tags.
>     *   All opening tags (e.g., `<|think|>, <search>, <python>`) must be paired with their exact corresponding closing tags (e.g., `</|think|>, </search>, </python>`).
>     *   Missing or redundant tags are prohibited. A tag block must be completely closed before a new tag block can be opened. Tags cannot be nested within one another, nor can their boundaries overlap.
>
> *   **$R\_{format} = 0$** if there are format violations, such as missing tags, unpaired tags, or incorrect tag sequences.
>
> We will explicitly add this formal definition in the revised manuscript to improve clarity.
>
> # Weakness 3
>
> We sincerely appreciate your constructive feedback. We completely agree that evaluating an agent's modeling proficiency based exclusively on the optimal objective value may be insufficient. Although matching the exact ground-truth objective remains the prevailing standard in recent literature (e.g., OptiMUS and OptMATH), we acknowledge the necessity of a more comprehensive, multi-dimensional assessment.
>
> - **Direct comparison with ground-truth formulations is highly challenging.** Because a single optimization problem can often be correctly formulated in several distinct ways, mathematically verifying the equivalence between different models is notoriously difficult.
> - **To address this, we now evaluate the structural alignment between LLM-generated models and the ground truths using key statistical features.** We report the proportion of models that match across metrics such as total variable count, binary/integer variables, number of constraints, and objective values. We must emphasize that these statistics serve purely as proxy indicators; a discrepancy in variable or constraint counts does not inherently mean the generated model is invalid.
>
> Crucially, our approach demonstrates the highest rate of structural correspondence with the ground-truth models, further validating its reliability. The results are in the OptMATH dataset.
>
> || Num Var Match | Num Bin Match | Num Cons Match |
> | :- | :- | :- | :- |
> | DeepSeek-V3   | 39.7 | 44.6| 28.9 |
> | GPT-4o| 27.1| 33.1| 24.6|
> | ORLM (8B) | 15.1| 12.7| 9.0|
> | LLMOPT (14B)  | 21.1| 19.2| 16.9|
> | OptMATH (32B) | 45.2 | 41.5| 31.9|
> | SIRL (32B)| 56.0 | 48.7| 50.0|
> | Ours (8B)| 57.8| 53.0| 53.6|

---

> > ### Author Rebuttal · Reviewer_43Fw · 2026-04-03
> >
> > Thanks for your response. My concerns have been solved and I will keep my positive score.

---

> > > ### Author Response · Authors · 2026-04-04
> > >
> > > Dear Reviewer 43Fw,
> > >
> > > Thanks for your valuable reviews. We are very grateful for your positive evaluation and support. We believe the manuscript has significantly improved thanks to your rigorous review and constructive suggestions. Thank you very much for your time and expertise.
> > >
> > > Best,
> > >
> > > Authors

---

### Official Review · Reviewer_yVCQ · 2026-03-12

**Soundness:** 2
**Presentation:** 3
**Significance:** 3
**Originality:** 3
**Overall Recommendation:** 4
**Confidence:** 3

**Summary:**

This paper proposes an agent, Opt-Miner, that autonomously identifies and learns missing knowledge from the web, thereby improving the agent's mathematical capabilities for optimization modeling. The authors design a tree-guided data synthesis pipeline and the R-GRPO algorithm for agent reinforcement learning. They compare the results of 18 agents and 6 datasets. Experimental results show that the performance of the trained 8B agent is comparable to that of 32B advanced commercial reasoning models.

**Compliance With Llm Reviewing Policy:**

Affirmed.

**Final Justification:**

During the rebuttal, the authors addressed my concern, so I maintain the positive score.

**Key Questions For Authors:**

Q1. Even when web search retrieves sufficient domain knowledge, are there cases where the model still fails to produce correct solutions (e.g., very low accuracy)? If so, what types of errors occur in these cases, and how do they reflect the limitations of the model’s mathematical reasoning ability?

Q2. In section 4.1.2, the paper claims that "Similarity identification" is done using an LLM-based classifier. What is the specific design of this classifier's prompts, and why is it designed this way? How is its robustness guaranteed across multiple iterations?


Q3. In lines 57-60, the paper argues that “Relying solely on final modeling accuracy fails to credit the quality of intermediate retrieval, leaving the agent unable to distinguish effective information seeking from redundant exploration.” However, its experimental evaluation still uses model accuracy as the metric. How does the proposed evaluation framework ensure a comprehensive assessment of the agent’s reasoning and information-seeking process? Beyond quantitative metrics, could qualitative evaluations (e.g., information completeness or logical consistency of the reasoning process) also be considered?

Q4. In section 4.2, lines 262-264 ("Hybrid Reward System" section), what is the “ground-truth evidence”? Is it already present in the original question, or is it human-annotated?

Q5. As for “Analysis on the Data Size”, does the data size increase refer to more samples within the same domain, or does it also involve expanding the coverage of different domains?

Q6. Table 6 in the appendix shows that other comparison datasets generally have a large proportion of manufacturing-domain tasks, whereas Opt-Miner-Bench does not. What is the reason for this? Should the proportion of each domain be kept equal among all datasets?

**Limitations:**

The authors did not discuss the limitations of their work. The reliance on web search substantially increases inference time. The authors can discuss the implications of this design choice for real-world deployment.

**Strengths And Weaknesses:**

**Strengths:**
- The paper identifies important limitations in current LLM-based mathematical modeling research, which is insightful. It highlights the limited real-world coverage of existing benchmarks and the lack of supervised signals for the information-seeking trajectory.
- The idea of transforming agents "from passive reasoners into active information seekers" is well-motivated and represents a good direction for improving mathematical reasoning capabilities.
- The experimental results are solid. With the R-GRPO algorithm for agent reinforcement learning, Opt-Miner-Qwen3-8B achieves performance comparable to that of the 32B agents and commercial reasoning models.
- Results of the ablation study show the importance of introducing web-search.

**Weaknesses:**
- The paper claims to construct a new benchmark, “Opt-Miner-Bench”, via a data synthesis process combined with human expert review. However, the sources and representativeness of this benchmark remain unclear. Appendix A.1 provides only a brief description, stating that the problems are “from both academic literature and industrial cases into a hierarchical tree structure,” but lacks crucial details. For example, it is unclear what types of papers or industry cases were used, what quality standards were adopted, and how the human review process was conducted. These details are essential for evaluating the benchmark's representativeness and understanding the practical effectiveness of the proposed method.
- When agents are equipped with web search capabilities, there is a potential risk that they may retrieve the original problems or their solutions from external sources. In such cases, the evaluation may reflect the ability to locate existing answers rather than true optimization modeling capability. It would be helpful for the authors to clarify how such data leakage is prevented.

---

> ### Author Rebuttal · Authors · 2026-03-30
>
> # Weakness 1
>
> We will detail the construction pipeline of Opt-Miner-Bench.
>
> - **Data Sources and Representativeness**. Seed problems were extracted from top OR journals (OR, MS, EJOR, and so on) using domain-specific keywords or adapted from real-world cases from the supply chain of an anonymous enterprise.
> - **Selection Criteria.** Problems selected based on three dimensions: broad domain coverage (covering all the provided domains), problem diversity (avoiding the same optimization models in the domain), and high complexity (at least 10 kinds of constraints).
> - We use LLM and human inspections to filter high-quality data.
>   - **LLM Difficulty Filtering**. We applied filters with models (GPT-4o and DeepSeek-V3) to retain only hard problems with less than 50% accuracy (with 5 attempts).
>   - **Human Validation Filtering**. OR experts manually inspected and corrected all surviving instances to ensure semantic coherence in the problem text and correctness of ground-truth Gurobi code.
>
> # Weakness 2
>
> Our data synthesis pipeline is to eliminate the risk of data leakage.
>
> - We do not use raw problems but use a data synthesis process. We either merge multiple distinct problems into a novel, complex scenario (Union) or transplant abstract mathematical logic into an entirely new industrial domain (Transfer).
>
> - As our synthesized problems are entirely new, their descriptions and corresponding optimization models do not exist on the web. Even though LLMs find part of the sources, there are still gaps for LLMs to totally solve the problem.
>
> # Question 1
>
> We analyzed these cases and categorized them into the following primary types:
>
> - **Code Errors**. The model cannot write executable solver code.
>
> - **Error Queries**. The model may propose incorrect queries that lead to not exactly relevant information.
>
> - **Retrieval Noise**. Retrieved documents often contain multiple problem variants. The model sometimes suffers from information overload, struggling to extract the exact information applicable to our specific scenario.
>
> - **Understanding Errors**. The model misunderstands the problem descriptions.
>
> Although we train the search and reasoning ability, our framework still cannot accurately guarantee sufficient domain knowledge.
>
> # Question 2
>
> The prompt for the similarity classifier is in **Appendix H.2**.
>
> - **Prompt Design**. The prompt instructs the LLM to output a strict binary score (0 or 1) by evaluating whether two HSTs share a common industrial domain, ignoring surface-level mathematical notations.
>
> - **Why this design?** Optimization problems are grounded in physical reality. Blindly merging completely unrelated domains (e.g., quantum computing and agriculture) creates absurd, logically incoherent scenarios. The similarity check is to ensure the synthesized scenario remains logically sound.
>
> - The **robustness** of this LLM-based classifier across multiple iterations is guaranteed through prompt designs:
>
>   - Instead of feeding raw, complex problem texts into the prompt, we feed it structured **HST representations** with separated secario contexts. This isolates these contexts from dense mathematical formulas, preventing the LLM classifier from being confused by mathematical notations across iterations.
>
>   - **Detailed Instructions**. The prompt includes detailed matching guidelines and examples, focusing on the scenario and entity nodes.
>
> # Question 3
>
> We introduced the **Retrieval Matching Score** (the F1 score of LLM queries and ground-truth evidence) reflecting the retrieval quality. In **Figure 3 (Middle)**, our R-GRPO significantly outperforms baselines, extracting effective information.
>
> |Method|Retrieval Matching Score|
> |-|-|
> |GRPO|0.01|
> |LLMOPT|0.008|
> |SIRL|0.01|
> |Ours|0.24|
>
> # Question 4
>
> The ground-truth evidence is used as the target retrieval keywords to construct the retrieval reward and supervise the agent's retrieval ability. The ground-truth evidence is recorded during our data synthesis pipeline. When we collect seed problems to build the base HSTs, we automatically extract and store the original search keywords and the explicit mathematical modeling techniques as the **ground-truth evidence set**. The data generation operators union these pieces of evidence from the original problems.
>
> # Question 5
>
> The increase in data size refers to adding more samples **proportionally across a fixed set of diverse domains**.
>
> # Question 6
>
> We ensure sufficient representative problems across major industrial domains and avoid the heavy domain skew (e.g., towards manufacturing). We think that if a benchmark is heavily dominated by a single domain, the overall macro-performance becomes biased. However, we cannot ensure an equal proportion of each domain, as some domains contain more kinds of common problems. We select the most representative problems for each domain.
>
> # Limitations
>
> Yes, we have made some efforts in the deployment. Please refer to our response to Weakness from cTSM.

---

> > ### Author Rebuttal · Reviewer_yVCQ · 2026-04-03
> >
> > Thanks for the detailed response.  The authors have addressed my concern. I will maintain the positive score.

---

> > > ### Author Response · Authors · 2026-04-04
> > >
> > > Dear Reviewer yVCQ,
> > >
> > > We thank the reviewer for their careful review and for the positive assessment of our work. We are glad that our responses have addressed your concerns. Thank you again for your support.
> > >
> > > Best,
> > >
> > > Authors

---

### Official Review · Reviewer_cTSM · 2026-03-13

**Soundness:** 2
**Presentation:** 3
**Significance:** 2
**Originality:** 2
**Overall Recommendation:** 5
**Confidence:** 2

**Summary:**

In the field of optimization modeling, this paper focus on the problem that existing models are often constrained by their static parametric knowledge and lack critical domain-specific expertise.
To overcome this, authors propose the Opt-Miner framework, which reformulates modeling problems into a Hierarchical Scenario Tree (HST) structure to embed scenario contexts and mathematical techniques within specific subtrees.
Besides, authors propose the Retrieval-based Group Relative Policy Optimization (R-GRPO), which is specifically designed to cultivate the agent's autonomous information-seeking capabilities.
Experimental evaluations demonstrate that Opt-Miner-Qwen3-8B achieves performance comparable to 32B state-of-the-art specialized agents.

**Compliance With Llm Reviewing Policy:**

Affirmed.

**Final Justification:**

Regarding network link latency: I am satisfied that the authors have provided details and empirical evidence, which has significant impact in real-world applications. I suggest that the authors add these details to the paper if it is accepted.

Regarding the contradiction between the retrieved documents and the model's internal priors: I now have a clearer understanding of the method to address it.

Overall, I will increase my score by 1.

**Key Questions For Authors:**

In real-world industrial settings, retrieved documents may occasionally contradict the model's internal priors. How does Opt-Miner resolve such information conflicts?

**Limitations:**

yes

**Strengths And Weaknesses:**

Strength:
Leveraging a hierarchical tree structure to foster an agent's ability to seek external information through web search is novel and innovative.

Weakness:
The inference process of Opt-Miner involves multiple rounds of iterative web-searching and document processing. During document retrieval, network link latency can lead to excessively slow response times, leading to slow  feedback.

---

> ### Author Rebuttal · Authors · 2026-03-30
>
> # Weakness
>
> We sincerely thank the reviewer for highlighting the latency. To mitigate this, we have made substantial engineering efforts.
>
> - **Engineering Efforts to Mitigate Latency**. To prevent excessively slow response times caused by network link latency or broken URLs, we implemented a robust retrieval pipeline:
>
>   - **Search Cache**. We implemented a caching mechanism for both search queries and parsed web pages. If the agent issues an identical query or accesses a previously processed URL, the system directly retrieves the results from the local cache. To avoid the explosion of the cache size, we set the maximum size of the cache and maintain the latest items.
>   - **Paralleled Document Retrieval**. We visit and process multiple retrieved web pages in parallel, significantly reducing the time of sequential visits.
>   - **Set the Timeouts.** We set a maximum timeout threshold for each network request (15 seconds). If a page fails to load within this limit, the agent skips it to maintain the overall workflow efficiency.
>   - **Retry Limits.** We restrict the maximum number of retry attempts for failed links (maximum 3 retries) to prevent the system from hanging on dead or extremely slow servers.
>
> - **Empirical Analysis of Latency**. We recorded the network statistics during our evaluation. The results are presented in **Table 1**.
>
>   **Table 1**: The network statistics.
>
> |Metric| Value|
> |-|-|
> |**Link Fetching Success Rate**| 93.3% |
> |**Average Retry Count per Query**| 0.14|
> |**Average Search Time per Problem**| 20.8|
> |**Average LLM Generation Time per Problem** | 25.8|
>
> - **The Trade-off of Time and Performance**. The overhead is highly acceptable. While web retrieval inevitably increases the latency, we observed a highly beneficial trade-off. When facing complex, knowledge-intensive optimization problems, baseline models lacking external knowledge often exhibit hallucination or repetitive reasoning steps. As shown in **Table 5** of our paper, the base Qwen3-8B takes 52.5 seconds on average. In contrast, by grounding its reasoning on explicitly retrieved documents, Opt-Miner reduces blind guessing, leading to a more concise and deterministic generation process. Consequently, the total inference time of Opt-Miner-Qwen3-8B is even slightly lower than that of the base model, while achieving an over 20% performance gain.
>
> # Question
>
> We think that the contradiction within the retrieved documents and the model's internal priors is inevitable. In fact, the retrieved documents may contain key points to solve the problem, or noises that are irrelevant. This is a fundamental question in retrieval-based methods. To address this challenge, we take two methods: (1) improving the quality of the retrieved documents to avoid the contradiction caused by the noise, and (2) improving the model's reasoning ability to effectively utilize the retrieved information.
>
> - **Improving the quality of retrieved information**.
>
>   - **Cross-Validation Across Multiple Documents**. When Opt-Miner performs a search, it does not rely on a single snippet. Instead, it retrieves multiple documents (e.g., the top-10) and implicitly performs cross-validation. It learns to identify the most consistently applied mathematical formulations across multiple credible sources.
>
>   - **Source Filtering via Retrieval Reward**. During our R-GRPO training, the retrieval matching reward explicitly supervises the agent to formulate precise, highly relevant search queries. By penalizing broad or off-topic searches, the agent learns to fetch high-quality, domain-specific technical documents, minimizing the retrieval of irrelevant or noisy information.
>
>
> - **Conflict Resolution via Prompt-Guided Thinking and Accuracy Reward**. When a genuine conflict arises between the model's generic internal prior and the specific retrieved industrial context, Opt-Miner resolves it through its explicitly trained reasoning process.
>
>   - Our prompt mandates the agent to engage in a rigorous thinking process to explicitly infer the most suitable reasoning path.
>   - More importantly, during the R-GRPO training, the final modeling accuracy reward acts as the ultimate supervisor. If the agent stubbornly relies on a flawed internal prior or retrieved documents that lead to failure, the agent will receive zero reward. Through extensive trial and error during RL training, the agent effectively learns that integrating the correct, context-specific external information yields higher rewards.
>
> - **Empirical Evidence on Conflict Resolution**. To explicitly quantify the impact of potential information conflicts, we analyzed the intersection of correctly solved instances with and without retrieval (Qwen2.5-4B-Instruct as the base model).
>
>   || Mamo ComplexLP | IndustryOR | OptMATH |
>   |-|-|-|-|
>   |Correct with both Retrieval and Non-Retrieval|56.4%|33.0%|27.7%|
>   |Correct ONLY with Retrieval|10.0%|8.0%|7.8%|
>   |Correct ONLY with Non-Retrieval|1.4%|2.0%|0.1%|

---

> > ### Author Rebuttal · Reviewer_cTSM · 2026-04-03
> >
> > Thanks to the authors for their rebuttal.
> >
> > Regarding network link latency: I am satisfied that the authors have provided details and empirical evidence, which has significant impact in real-world applications. I suggest that the authors add these details to the paper if it is accepted.
> >
> > Regarding the contradiction between the retrieved documents and the model's internal priors: I now have a clearer understanding of the method to tackle it.
> >
> > Overall, I will increase my score by 1.

---

> > > ### Author Response · Authors · 2026-04-04
> > >
> > > Dear Reviewer cTSM,
> > >
> > > We sincerely thank you for the positive feedback and for the time spent evaluating our revisions. We are pleased to hear that our responses have addressed your concerns. We will follow your suggestions and add the details during the rebuttal to the paper.
> > >
> > > Best,
> > >
> > > Authors

---

### Decision · Program_Chairs · 2026-04-30

**Decision:**

Accept (regular)

**Comment:**

Reviewers appreciated the "novel and innovative" use of Hierarchical Scenario Tree structures to structure the search for mathematical modelling, noting that it addresses a hole in the literature. They found the introduced version of GRPO well suited for the task and the empical justification for its effectiveness strong, hence making a solid contribution to the conference. Please encorporate the changes suggested by the reviewers for the camera-ready version.